# Nanostructure and Luminescent Properties of Bimetallic Lanthanide Eu/Gd, Tb/Gd and Eu/Tb Coordination Polymers

Helena Brunckova [1,*], Erika Mudra [1], Lucas Rocha [2], Eduardo Nassar [2], Willian Nascimento [2], Hristo Kolev [3], Maksym Lisnichuk [1], Alexandra Kovalcikova [1], Zuzana Molcanova [1], Magdalena Strečkova [1] and Lubomir Medvecky [1]

[1] Institute of Materials Research, Slovak Academy of Sciences, Watsonova 47, 040 01 Kosice, Slovakia; emudra@saske.sk (E.M.); mlisnichuk@saske.sk (M.L.); akovalcikova@saske.sk (A.K.); molcanova@saske.sk (Z.M.); mstreckova@saske.sk (M.S.); lmedvecky@saske.sk (L.M.)

[2] Universidade de Franca, Science &Technology, Av. Armando Salles Oliveira 201, Franca 14404-600, Brazil; lucas.rocha@unifran.edu.br (L.R.); eduardo.nassar@unifran.edu.br (E.N.); willianenmelo@outlook.com (W.N.)

[3] Institute of Catalysis, Bulgarian Academy of Sciences, Acad. G. Bonchev St., BU-1113 Sofia, Bulgaria; hgkolev@gmail.com

* Correspondence: hbrunckova@saske.sk; Tel.: +421-55-7922-455

**Abstract:** This study presents the synthesis, structural and luminescence properties for lanthanide metal–organic frameworks (LnMOFs), which belong to the sub-class of coordination polymers. The series of nanosized LnMOFs (Ln = Eu, Gd, Tb, $Eu_{0.5}Gd_{0.5}$, $Tb_{0.5}Gd_{0.5}$ and $Eu_{0.5}Tb_{0.5}$) was prepared by solvothermal synthesis using a modulator (sodium acetate). We investigated the various surface chemistry compositions of the isostructural LnMOFs with a [Ln(btc)] structure (BTC: Benzene-1,3,5-tricarboxylate) by X-ray photoelectron spectroscopy (XPS). The XPS confirmed the mixed-valent $Eu^{3+}$ and $Eu^{2+}$ compounds, and the presence of Tb in both +3 and +4 valence states, and one +3 valency of Gd. A nanostructure of mixed LnMOFs (EuGd, TbGd and EuTb) with a rod-like shape is related to luminescence properties. The MOFs (EuTb and EuGd) presented Comission Internationale de l'Éclairage (CIE) chromaticities of x = 0.666 and y = 0.331, and x = 0.654 and y = 0.348, respectively, in the red region. They were better than the values desired for use in commercial phosphors, which are x = 0.64 and y = 0.35. For [Tb/Gd(btc)], the CIE coordinates were x = 0.334 and y = 0.562, presenting emissions in the green region. Bimetallic LnMOFs are very promising UV light sensors for biological applications.

**Keywords:** coordination polymers; metal-organic frameworks; lanthanides; solvothermal synthesis; X-ray photoelectron spectroscopy; luminescence

## 1. Introduction

Coordination polymers (CPs) are constructed of metal ions and bridging ligands that combine them into solid-state structures extending in one (1D), two (2D), or three dimensions (3D). Two- and three-dimensional CPs with potential voids are often designated as metal–organic frameworks (MOFs) [1–5]. MOFs are a well-recognized sort of attractive blended materials built from varied metal centers and multidentate organic ligands via coordination bonds [6]. MOFs are a remarkable type of porous materials because of their potential in many uses, such as gas absorption, molecular segregation, storage, optics and catalysts [7,8]. Lanthanide metal–organic frames (LnMOF) are recognized as specific types due to their special coordination characteristics and excellent optical properties resulting from 4f electrons [9]. LnMOFs are well known for their great Stokes' shifts, high color purity, and rather prolonged luminescence lifetimes outgoing from f-f transitions trough the "antenna effect" [9,10]. LnMOFs display sharp and intense luminescence, emissions in the primal color scale (red, green and blue) that entirely cover the whole visible spectrum and therefore can be well combined for the design of white-light emitting materials [11]. Luminescent LnMOFs are now of great benefit and significance due their

technical importance as sensors [6]. Trivalent lanthanides ($Ln^{3+}$) are an ideal selection, as the inorganic metal cation in the construction of nano-MOFs and Ln centers allows for multimodal imaging [12]. The trivalent ions of europium and terbium radiate intensive red and green light, respectively, while the $Gd^{3+}$ complex emits in the blue zone due to its high lowest level of emitting [10,13]. As a result, varied lanthanide complexes ($Gd^{3+}$, $Tb^{3+}$ and $Eu^{3+}$) possess eventual applications in the creation of white-light emission [10].

Lanthanide MOFs built using coordination bonds among Ln ions and organic ligands are hopeful materials due to their porous crystalline structures, rich mixtures and simple preparation [14,15]. The set of LnMOFs [Ln(btc)] was prepared by solvothermal synthesis via classical $H_3BTC$, and was used for the luminescence sensing of benzene homologue solutions [16]. [Eu(btc)] and [Tb(btc)] show acute and stark emissions in the visible light range. [Eu(btc)] was prepared by solvothermal synthesis using N, N-dimethylformamide (DMF) as the solvent, with modulator sodium acetate (NaOAc) [6,16,17]. In an effort to optimize the preparation requirements, the luminescent properties of [Eu(btc)($H_2O$)(dmf)] with different morphologies, which show similar red light emissions and different intensities of short ultraviolet radiation, were investigated [6]. Ren et al. [18] studied the influence of the MOF size as sensors, and prepared bulk TbMOF and distinct sizes of nano-TbMOFs by the ultrasound-supported method via NaOH as a modulating agent, and demonstrated that these [Tb(btc)] have the same structure. Bimetallic MOF has two luminescence centers into this network, similarly to [Eu/Tb/(btc)] [17], [Eu/Gd(btc)] and [Tb/Gd(btc)] [10,12]. The [Eu/Tb/(btc)] emission color is orange–red [17] and Eu/GdMOF results in strong red emissions [19]. The [Ln(btc)] sets were built with trivalent lanthanide ($Ce^{3+}$, $Y^{3+}$, or a combination thereof) [8]. Heterometallic Eu/Tb-MOF exhibits luminescence, and was obtained by solvothermal synthesis with various ligands, such as 2-phenylsuccinate (psa) [20], 4,4'-oxybis(benzoate) acid ($H_2OBA$) [21], furan-2,5-dicarboxylic acid ($H_2FDA$) [22] and 1,3-bis (3,5-dicarboxyphenyl) imidazolium chloride ($H_4L^+Cl^-$) [23]. A series of hybrid LnMOFs as [$Eu_xGd_{1-x}$(ndc)] (x = 0.0005 − 0.01; $H_2NDC$ = 1,4-naphthalene dicarboxylic acid) using aback power transmission were fabricated for the physiologic sensing of the temperature [24].

The aim of the work was to characterize the chemical content, binding structures and morphologies on the surface of the LnMOF series (Ln = Eu, Gd, Tb) and mixed (Ln = $Eu_{0.5}Gd_{0.5}$, $Tb_{0.5}Gd_{0.5}$ and $Eu_{0.5}Tb_{0.5}$) prepared by solvothermal synthesis using $DMF/H_2O$ solution and adding sodium acetate as modulator. Moreover, the effect of mixed different bimetallic lanthanides in [Ln(btc)(dmf)($H_2O$)] nanostructures on the surface chemistry composition was studied. The novelty of this research is the TEM characterization of different nanostructures of bimetallic LnMOFs. The luminescence property of fabricated bimetallic LnMOFs were explored. Luminescence spectra presented high excitation bands and intense emissions. The resulting luminescent LnMOFs can be applied in the production of various optical equipment.

## 2. Materials and Methods

### 2.1. Materials and Chemicals

All of the chemical agents and dissolvents were purchased from Sigma-Aldrich at analytical grade and applied without another purgation. Europium(III) nitrate hydrate $Eu(NO_3)_3 \cdot 5H_2O$, gadolinium(III) nitrate hydrate $Gd(NO_3)_3 \cdot 6H_2O$, terbium(III) nitrate hydrate $Tb(NO_3)_3 \cdot 6H_2O$, N,N-dimethylformamide (DMF), 1,3,5-benzenetricarboxylic acid (BTC), sodium acetate (NaOAc) and deionized water were availed to the solvothermal synthesis.

### 2.2. Preparation of the LnMOFs

The LnMOFs were prepared via modified solvothermal synthesis [6,16,17] pursuant to the previous work [25]. Lanthanide(III) nitrate hydrate $Ln(NO_3)_3 \cdot xH_2O$ (1.0 mmol) and $H_3BTC$ (0.21 g, 1.0 mmol) were dissolved in a 30 mL mixture of $DMF/H_2O$ (1:1 v/v) solvents (Ln = Eu, Gd, Tb) together with the modulator NaOAc (0.3 mmol). The preparation procedures for the Eu, Gd and Tb lanthanide MOFs were the same, and were performed

using different starting nitrates using $Eu(NO_3)_3 \cdot 5H_2O$ (0.443 g), $Gd(NO_3)_3 \cdot 6H_2O$ (0.448 g) and $Tb(NO_3)_3 \cdot 5H_2O$ (0.449 g). The ratio of Ln:TBC was 0.36. The three solutions of [Eu(btc)] (Eu-1), [Gd(btc)] (Gd-2) and [Tb(btc)] (Tb-3) were mixed at 25 °C for 1 h and heated at 60 °C for 48 h, and then cooled to room temperature to give white (Eu-1 and Gd-2) and colorless (Tb-3) crystals. After the synthesis, the products were isolated by centrifugation and washed several times with ethanol and water, respectively, and then dried in air. The prepared Eu-1, Gd-2 and Tb-3 resulted in a yield of 66% (0.319 g), 69% (0.270 g) and 70% (0.339 g), respectively, without elemental analysis. The preparation of the mixed bimetallic $[Eu_{0.5}Gd_{0.5}(btc)]$ (EuGd-4), $[Tb_{0.5}Gd_{0.5}(btc)]$ (TbGd-5) and $Eu_{0.5}Tb_{0.5}(btc)$ (EuTb-6) was the same as that for simple [Ln(btc)], only the pure Ln nitrate was exchanged via a mixture of two corresponding nitrates. For the preparation of the EuGd-4 powder, the nitrate of Eu (0.223 g, 0.5 mmol) and Gd (0.224 g, 0.5 mmol) with BTC (0.21 g, 1.0 mmol) and NaOAc (0.03 g, 0.3 mmol) was dissolved in 30 mL $DMF/H_2O$. For TbGd-5 or EuTb-6 syntheses were used $Tb(NO_3)_3 \cdot 6H_2O$ (0.225 g, 0.5 mmol) separately. The bimetallic LnMOF powders were formed after heating at 60 °C for 48 h. The preparation procedures for the other lanthanide MOFs were analogous. The experimental synthesis of each sample was repeated three times. The syntheses of Tb-3 and TbGd-5 MOFs were performed without the NaOAc modulator for the SEM and TEM analysis.

The comparison with the other methods and procedures for solvothermal synthesis of LnBTC (Ln = $Eu_{0.5}/Gd_{0.5}$ or $Tb_{0.5}/Gd_{0.5}$) used the selected lanthanide chloride salts, sodium trifluoroacetate (NaTFA), and BTC in a ~1:0.9:0.6 Ln:TFA:BTC ratio, with the solvents water and DMF [12] and a ball milling preparation of $[Eu_{0.5}/Gd_{0.5}(btc)]$ or $[Tb_{0.5}/Gd_{0.5}(btc)]$ with the $H_3BTC$ and the respective lanthanide carbonate hydrates $Ln_2(CO_3)_3 \cdot xH_2O$ in a 2:1 molar ratio. The other solvothermal syntheses of LnMOFs have been reported with various ligands, such as psa [20], $H_2OBA$ [21], $H_2FDA$ acid [22], $H_4L^+Cl^-$ [23] and NDC [24].

### 2.3. Characterization of the LnMOFs

The FTIR analysis of the LnMOF compounds was performed using the FTIR spectrometer Shimadzu IRAffinity1 (KBr pellets). The composition and structure of the samples were obtained using an X-ray diffraction (XRD) instrument (X' Pert Pro, Philips, Amsterdam, The Netherlands) via Cu $K_\alpha$ radiation. The surface morphologies were characterized using scanning electron microscopy (SEM) (Auriga Compact, Carl Zeiss, Jena, Germany) and high resolution transmission electron microscopy (TEM), (JEOL-JEM 2100F), using a scanning transmission electron microscope (STEM) and energy dispersive X-ray (EDS, Oxford Energy TEM 250) spectroscopy. The additional composition and valence state inquiry were obtained by X-ray photoelectron spectroscopy (XPS). The energy range was scaled using a standardizing C 1s line of acquired hydrocarbons to 285.0 eV for the electrostatical sample. The luminescence spectra were characterized at room temperature, using a continuous Xenon lamp or a flash Xenon lamp in a Horiba Jobin Yvon Fluorolog-3 spectrofluorimeter equipped with a double excitation monochromator and a single emission monochromator. The excitation and emission spectra were obtained at various emission and excitation wavelengths, respectively.

## 3. Results and Discussion

### 3.1. Structural Characterization of the LnMOFs

The TG and DSC curves of LnMOFs (Eu-1, Gd-2, Tb-3) prepared by sovothermal synthesis using the modulator (NaOc) are shown in Figure 1. The TG curves are similar, and all of them display a two-step or three-step weight loss [6]. The initial weight loss—starting at around 100 °C and continuing up to 160 °C—observed for all of the samples can be ascribed to the loss of coordinated solvent molecules (DMF and $H_2O$) [25]. The TG curves of Eu-1and Gd-2 show the two main steps of gradual weight loss process before 220 °C, attributed to the release of $H_2O$ and two DMF molecules, and 7.0% in 220–410 °C temperature range, corresponding to the loss of coordinated DMF molecules. Above 410 °C, the complete framework sequentially begins to collapse upon further heating. All

of the samples had similar thermal stability up to 540–600 °C. The weight loss at higher temperatures of 540–600 °C was attributed to the thermal decomposition of [Ln(BTC)] to Ln oxides. The DSC curves result show that the end point of the endo peak depends on chemical component of the lanthanides (Eu-1: 620 °C, Gd-2: 658 °C and Tb-3: 673 °C).

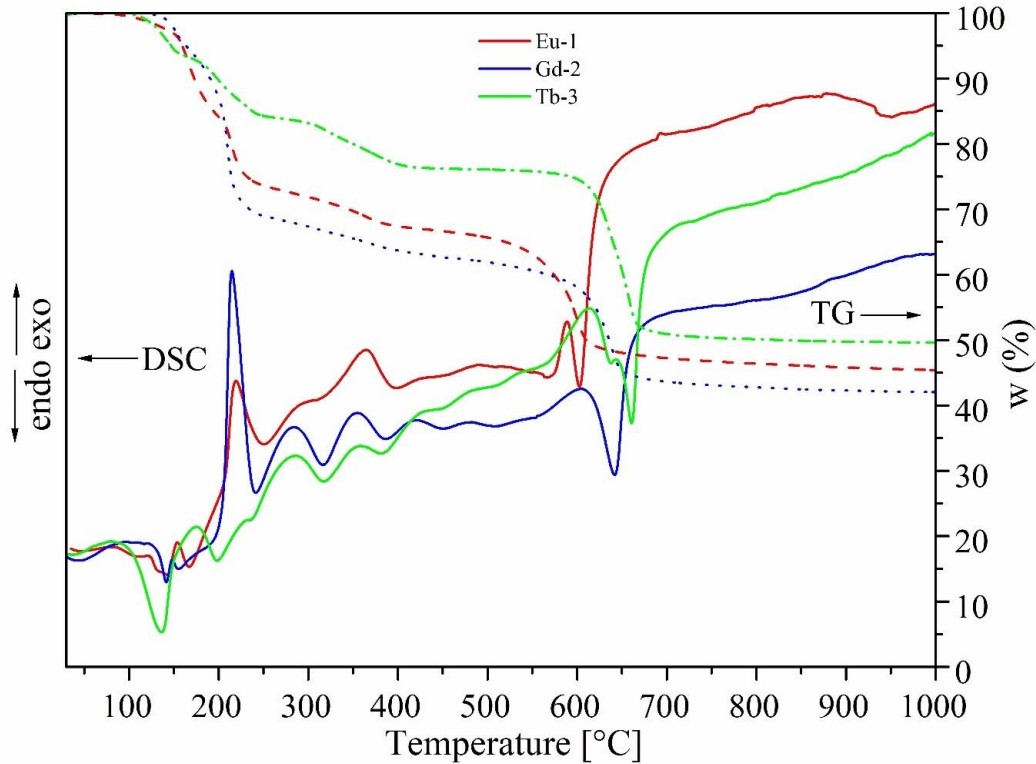

**Figure 1.** DSC/TG curves of LnMOF (Ln = Eu, Gd, Tb) prepared by solvothermal synthesis.

Figure S1 presents the FTIR spectra of LnMOF isostructural compounds prepared using solvothermal synthesis [25]. The wide peak at 3420 cm$^{-1}$ is assigned to ν(O-H) groups. The effect of acetate groups from sodium acetate can be noticed in the regions at 2983, 2786 and 2441 cm$^{-1}$, which are assigned to stretching ν(C-H) vibrations. In the spectrum of H$_3$BTC acid, the characteristic bands of the carboxyl group of BTC are at 3090, 1720, and 537 cm$^{-1}$. In the spectra of MOFs, no band at 1720 cm$^{-1}$ corresponding to the COOH groups can be seen, designating the complete deprotonation of the carboxylic acid and coordination of COO$^-$ groups to the lanthanide centre. The peak at 1678 cm$^{-1}$ belongs to ν (C=O) of DMF. In the spectra, the bands in the zones of 1568–1538 cm$^{-1}$ and 1386 cm$^{-1}$ were marked as stretching vibrations of the COO$^-$ groups ν$_{as}$ and ν$_s$, respectively. The powerful peaks provide the C-H bending benzene vibrations that shifted to the region of 775 and 710 cm$^{-1}$ [8,26]. The peak which appeared at 563 cm$^{-1}$ can be assigned to the stretching vibration of Eu, Gd and Tb-O [1,27]. The structural designation of the LnMOFs is marked as [Ln(btc)(H$_2$O)(dmf)] [6,9,25].

The XRD diffractograms of the samples are shown in Figure S2. The XRD peaks are in good compliance with previous report indicating that the obtained MOFs are isostructural [25]. The XRD diffractograms are in good agreement with patterns of [Ln(btc)], which have a tetragonal structure, as [La(btc)] simulated [28]. The diffraction patterns indicate the formation of [Ln(btc)], (Ln = Eu, Gd, Tb, Eu$_{0.5}$Gd$_{0.5}$, Tb$_{0.5}$Gd$_{0.5}$ and Eu$_{0.5}$Tb$_{0.5}$) in 2θ of 10.5°, 18.3°, 20.3°, 27.5° and 32° of the most important peaks [25,28]. The other peaks also match the [La(btc)] structure. The [Eu(btc)(H$_2$O)(dmf)], built with a Eu$^{3+}$ cation and the organic linker BTC via coordination bonding, is a 3D framework with a tetragonal structure [6,9,16]. XRD showed that the [Eu(btc)(H$_2$O)(dmf)] structure is not changed using a slight addition of NaOAc, but it will be transformed to a monoclinic structure with the effect of a major quantity of modulator agent [6,8].

### 3.2. XPS Characterization on the Surface of the LnMOF

The XPS survey spectra of the samples are presented in Figure S3, with all of the peaks congruent to the typical electronic Eu, Gd, Tb, C, O and N transitions. The total concentration of the elements in the atomic percentage on the surface of the powders was obtained. The composition of the powders on the surface was determined from the intensities of the signals, and is summarized in Table S1. The relative atomic concentration of Eu, Gd and Tb on the surface of the Eu-1 (5.9 at.%), Gd-2 (4.1 at.%) and Tb-3 (3.8 at.%) MOFs was higher than mixed EuGd-4 (2.5 at.%), TbGd-5 (1.9 at.%) and EuTb (2.0 at.%), respectively. The surface composition for EuTb-6 in at.% of Eu, Tb, C, O and N was determined as 1.8, 2.0, 53.7, 33.8 and 8.7, respectively. The values calculated for the assumed formulae of $[Ln(btc)(dmf)_2(H_2O)]$, $\{Ln[C_6H_3(CO_2)_3](C_3H_7NO)_2(H_2O)$, and $C_{15}H_{19}N_2O_9Ln$ are $C_7H_xN_2O_7Eu$, $C_{11.6}H_xN_{2.8}O_9Gd$, $C_{12.4}H_xN_{3.5}O_{9.4}Tb$, $C_{10.8}H_xN_{1.4}O_{7.6}$ $Eu_{0.52}Gd_{0.48}$, $C_{14.1}H_xN_{2.6}O_{10.6}$ $Tb_{0.47}Gd_{0.53}$ and $C_{14.1}H_xN_{2.3}O_{8.9}$ $Eu_{0.47}Tb_{0.53}$.

In Figure 2, the high resolution HR XPS Eu 3d, Gd 3d, Tb 3d and C 1s, O 1s and N 1s spectra of [Ln(btc)] powders can be observed. The XPS Eu 3d spectrum of Eu-1, EuGd-4 and EuTb-6 contains two significant peaks at 1135 eV ($Eu^{3+}$ $3d_{5/2}$) and 1165 eV ($Eu^{3+}$ $3d_{3/2}$) binding energies, together with two minor satellites that appear at 1128 eV ($Eu^{2+}$ $3d_{5/2}$) and 1155 eV ($Eu^{2+}$ $3d_{3/2}$) [10,29]. The XPS spectrum of Gd-2, EuGd-4 and Tb-Gd-5 shows peaks centered at 1088 eV (Gd $3d_{5/2}$) and 1220 eV (Gd $3d_{3/2}$), with one (+3) valence state of gadolinium. The XPS Tb 3d spectrum for the Tb-3, TbGd-5 and EuTb-6 samples represents two peaks at 1242 and 1277 eV, assigned to $3d_{5/2}$ and $3d_{3/2}$ of $Tb^{3+}$, respectively. The presence of $Tb^{4+}$ was found in a small peak at 1250 eV [10,14,30,31]. XPS corroborated the coordination impact among europium, gadolinium and terbium ions and the BTC ligand. The XPS spectrum of O 1s, C 1s and N 1s in all of the LnMOF powders showed peaks centered at 532, 285 and 402–407 eV, respectively.

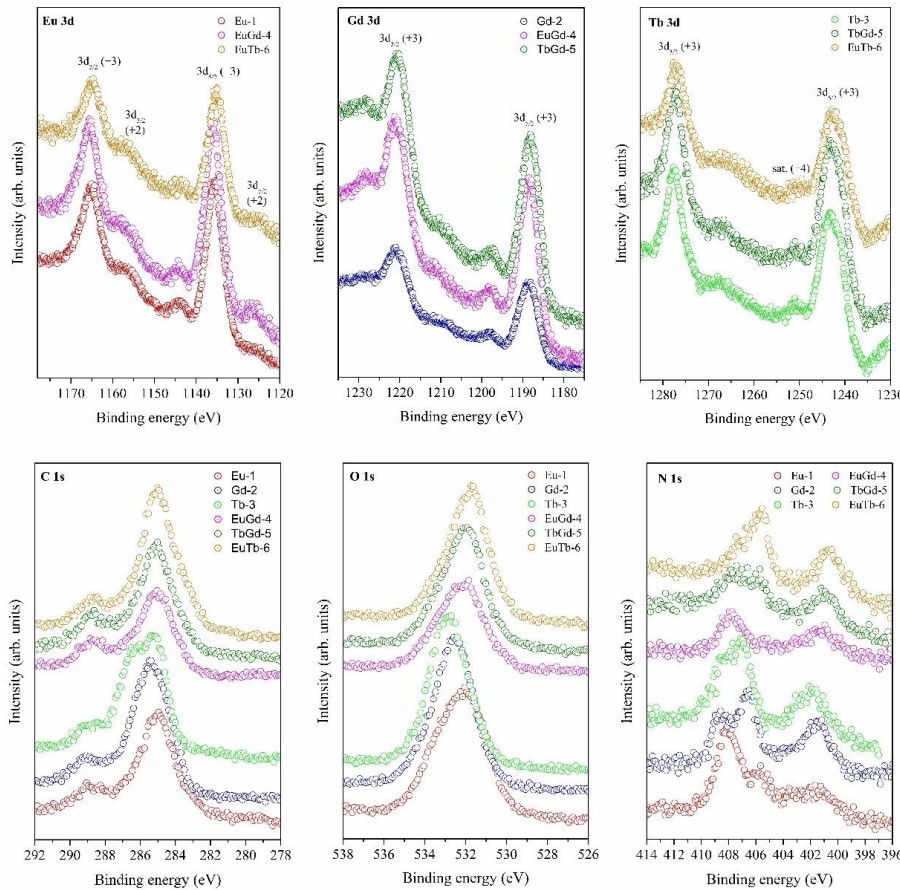

**Figure 2.** HR XPS spectra of the Eu 3d, Gd 3d, Tb 3d, C 1s, O 1s and N 1s of [Ln(btc)] (Eu-1, Gd-2, Tb-3, EuGd-4, TbGd-5 and EuTb-6) powders.

In Figure S4, XPS presents evidence of the attendance of two valency states for Eu ($Eu^{3+}/Eu^{2+}$) and Tb ($Tb^{3+}/Tb^{4+}$) at the surface of the MOFs. In the [Eu(btc)] and [$Eu_{0.5}Tb_{0.5}$(btc)] (MOFs), the $Eu^{3+}$ and $Eu^{2+}$ valency states were noticed at 135 and 130 eV, respectively. The states of $Tb^{3+}$ and $Tb^{4+}$ in [Tb(btc)], [$Tb_{0.5}Gd_{0.5}$(btc)] and [$Eu_{0.5}Tb_{0.5}$(btc)] were confirmed. The spectrum of Tb 4d acquired for the Tb-3 and EuTb-6 sample shows a strong peak at 150 eV and a small peak at 155 eV, attributed $Tb^{3+}$ and $Tb^{4+}$, respectively. The $Eu^{2+}$ concentration of Eu-1 and EuTb-6 was detected to be 0.8 and 0.1 at.%. In the Tb-3 and EuTb-6 samples, the $Tb^{4+}$ amount was determined to be 0.2 and 0.1 at.%. The presence of the two valence states, $Eu^{3+}/Eu^{2+}$ and $Tb^{3+}/Tb^{4+}$, in the corresponding samples was confirmed. This could also be caused by the preparation of samples and drying in air.

The XPS core-level spectra of the O 1s, C 1s and N 1s of Eu-1 and EuTb-6 are shown in Figure 3. The total atomic percentage concentrations of C, O and N in the EuTb-6 powder on the surface were determined to be 53.7, 33.8 and 8.7 %, respectively. The C 1s peak for EuTb-6 can be sectioned into three peaks at 285.0, 286.8 and 288.9 eV, which were assigned to C-H, C-C (35.4 at.%), C-O, C-O-H (8.2 at.%) and C=O. C-O-C (10.1 at.%), respectively [32,33]. The O 1s peak can be composed of three peaks at 531.3, 531.8 and 533.1 eV, which were ascribed to C-O (8.8 at.%), H-C-O (17.2 at.%) and O-H, with Ln-C-O (7.8 at.%) bindings of EuTb-6 [33,34]. The N 1s peak of the EuTb-6 sample, as shown in Figure 3, contains three peaks assigned to the subsequent bands: 400.6 eV (N-H-N-O, C-H-OH), 405.6 eV (N-O), and 407.2 eV (O-N-C-H) [34,35]. The XPS results for C 1s and N 1s are in good agreement with the FTIR ν(C=O) and δ (O=C-N) from coordinated DMF, respectively.

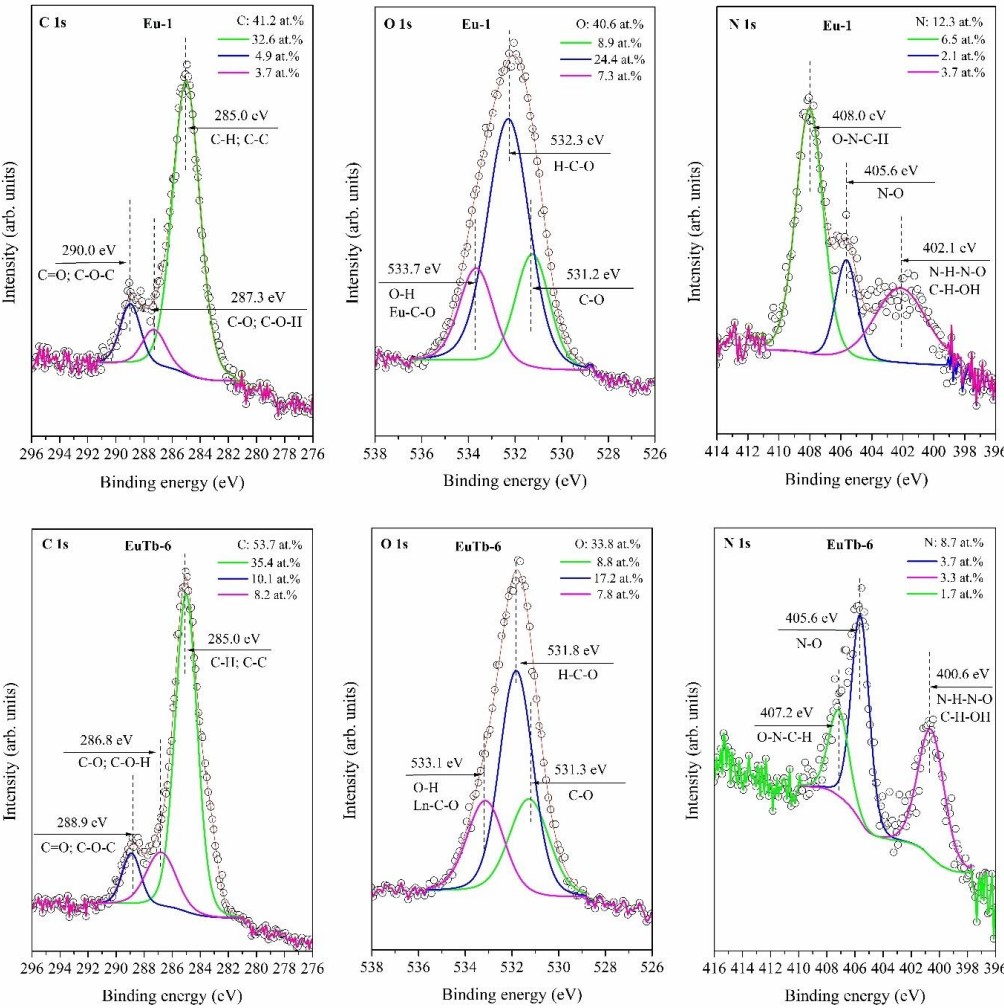

**Figure 3.** HR XPS spectra of C 1s, O 1s and N 1s, and curve-fitted peaks for [Eu(btc)] and [$Eu_{0.5}Tb_{0.5}$(btc)] powders.

### 3.3. Surface Morphologies of LnMOFs

The surface morphology of the as-synthesized powders was characterized using SEM and TEM. The images of the SEM and EDS analysis of LnMOF in Tb-3, TbGd-5 and EuTb-6 (insert TEM images) are shown in Figure S5 and Figure 4. The TbMOF morphology formed typical rods in the range of 2 to 10 μm without a modulator (Figure S5a [15]. In Figure S5b, needle-shaped nanocrystals of TbMOF after the addition of the modulator NaOAc can be observed [18]. The EDS spectra (Figure S5c,d) of [Tb(btc)] disclosed the presence of C, O, N and Tb elements in at.% [36]. The EDS spectra of [Tb(btc)] confirmed that the large rods and nano-rods possess equal molar ratios of C, O, N and Tb [6]. Similarly, the crystals of the TbGd-5 (Figure 4a) are evenly spaced, and form pillar-like rods with a size of 10–30 μm (without a modulator). In the EuTb-6 sample (synthetized using the modulator NaOAc), well-distributed nano-rods were observed (Figure 4c). The EDS spectra are shown in Figure 4b,d. The elements are compliant with the composition of the required LnMOF {[$Tb_{0.5}Gd_{0.5}$(btc)] and [$Eu_{0.5}b_{0.5}$(btc)]} structures.

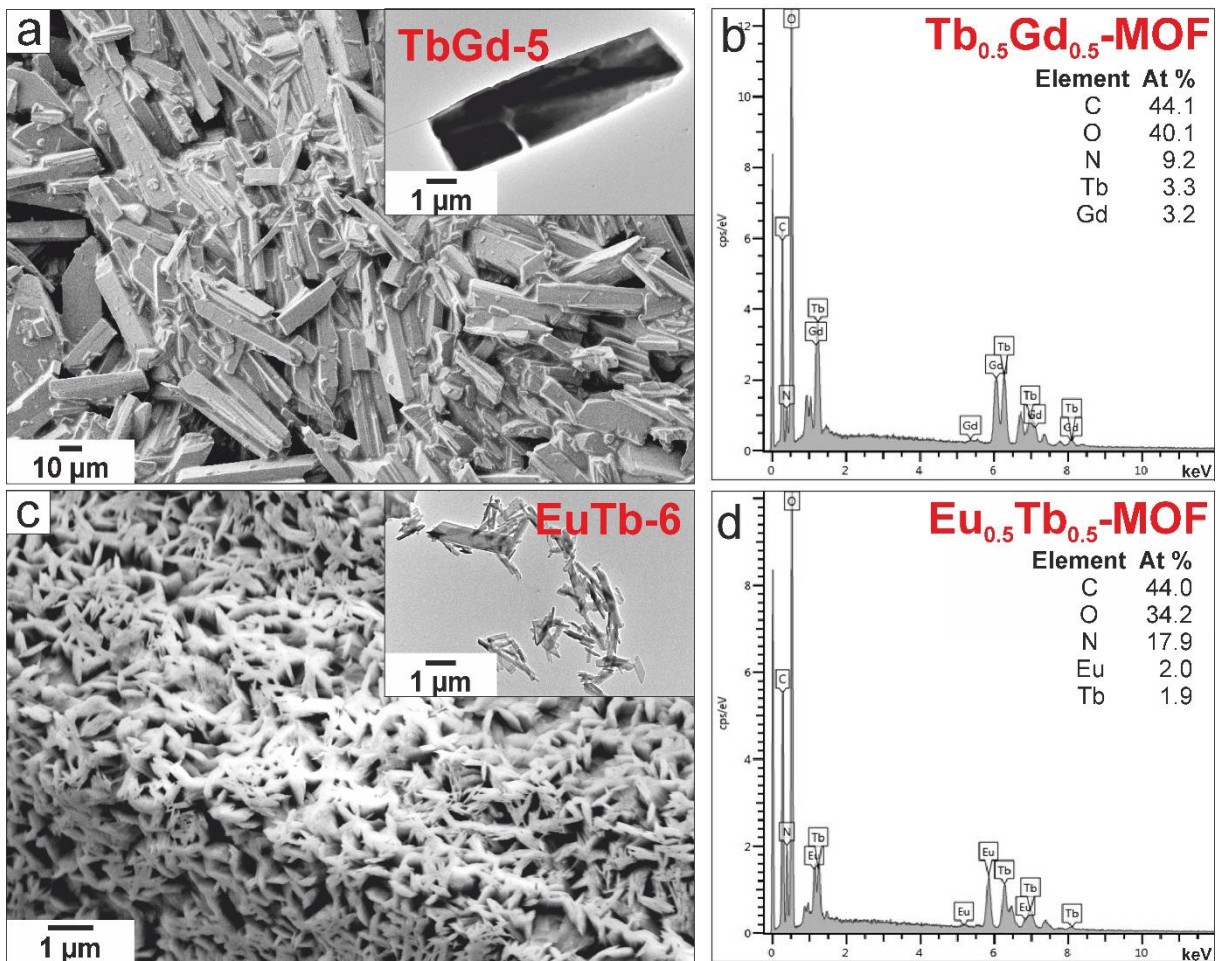

**Figure 4.** SEM morphology and TEM images (in insert) of LnMOF powders prepared by the solvothermal synthesis of (**a**) TbGd-5 (DMF/$H_2O$) and (**c**) EuTb-6 (DMF/$H_2O$/NaOAc); and the EDS spectra of (**b**) [$Tb_{0.5}Gd_{0.5}$(btc)] and (**d**) [$Eu_{0.5}Tb_{0.5}$(btc)].

The TEM micrographs of the Eu-1, Gd-2 and Tb-3 samples and the hybrid MOF (EuGd-4, TbGd-5, and EuTb-6) are displayed in Figure 5. It is absorbing to comment that all of the LnMOF samples prepared using the modulator NaOc exhibit a uniform morphology of their rods, with a size of 100–400 nm [11]. Figure 5d presents one rod with a length of 850 nm and a width of 400 nm, within which small nanorods (length 300–450 nm and width 50–100 nm) are incorporated. In Table 1, the variation of the lanthanide in the LnMOFs results in quite uniform rod-like nanocrystals with a length and width of

850 nm and 400 nm, respectively (EuGd-4), included smaller particles with a length of 300 nm and a width of 50 nm. The size of the nanorods decreases in this order: EuGd-4, TbGd-5 and EuTb-6. The records acquired using TEM were fully alike and agreed well with the results of samples prepared using solvothermal synthesis with the addition of a modulator (NaOAc) gained from SEM. The various SEM and TEM morphologies of the bimetallic [Ln(btc)] ($Eu_{0.5}Gd_{0.5}$, $Gd_{0.5}Tb_{0.5}$ and $Eu_{0.5}Tb_{0.5}$) were changed depending on the synthesis chemistry composition of the $Ln^{3+}$ ions from the nanorods (100–300 nm) for [$Eu_{0.5}Gd_{0.5}$(btc)] and [$Gd_{0.5}Tb_{0.5}$(btc)] to nanoparticles (20–100 nm) for [$Eu_{0.5}Tb_{0.5}$(BTC)].

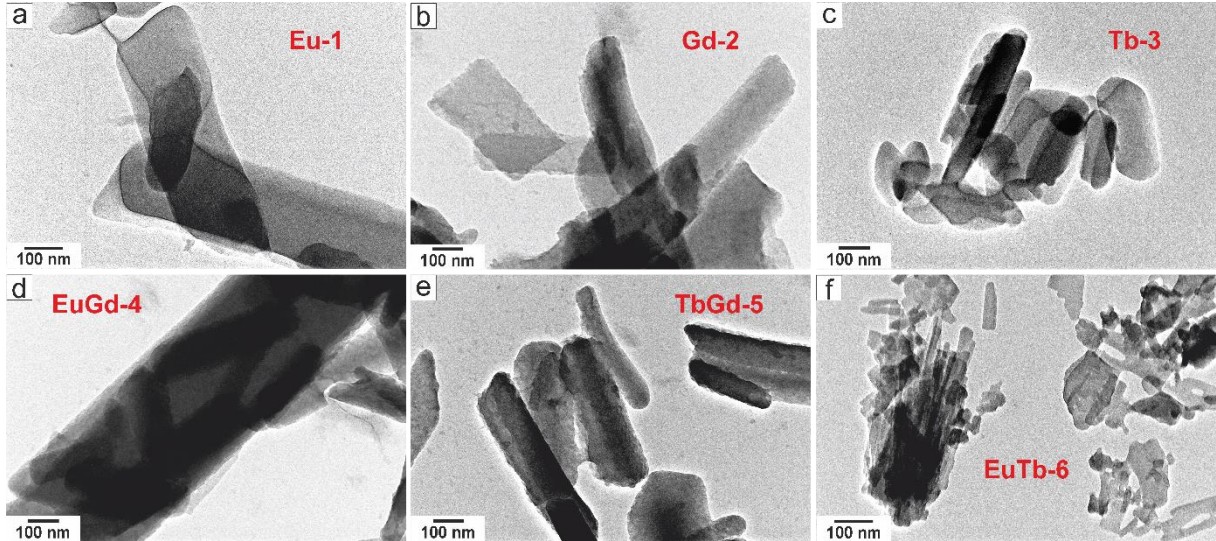

**Figure 5.** TEM images of the LnMOF powders: (**a**) Eu-1, (**b**) Gd-2, (**c**) Tb-3, (**d**) EuGd-4, (**e**) TbGd-5 and (**f**) EuTb-6.

**Table 1.** The variation in sizes of the nanorods obtained from the TEM images (Figure 5).

| LnMOF | Size of Nanorod (nm) | | Size of Included Particles (nm) | |
|---|---|---|---|---|
| | **Length** | **Width** | **Length** | **Width** |
| Eu-1 | 820 | 280 | 350 | 100 |
| | 600 | 150 | 200 | 50 |
| Gd-2 | 730 | 150 | 150 | 50 |
| | 600 | 150 | 100 | 40 |
| Tb-3 | 400 | 50 | 150 | 60 |
| | 250 | 50 | 100 | 50 |
| EuGd-4 | 850 | 400 | 450 | 100 |
| | | | 300 | 50 |
| TbGd-5 | 400 | 150 | 150 | 30 |
| | 300 | 30 | 130 | 20 |
| EuTb-6 | 140 | 20 | 40 | 20 |
| | 110 | 20 | 30 | 20 |

The STEM and EDS mapping of the elements in the LnMOFs (Figure 6) elucidate the homogeneous partition of the Ln, O and C elements through the surface of the samples [37]. The STEM images and EDS mapping (Figure 6a) of TbMOF further detected the presence of Tb, O and C as incorporated elements in the respective particles [35]. In Figure 6b–d, at different magnifications, are shown the nanorods of mixed EuGd-4, TbGd-5 and EuTb-6, respectively. The concentration of the elements (in at.%) from the EDS spectra and the mapping of the Tb-3, EuGd-4, TbGd-5 and EuTb-6 samples from the STEM/EDS are summarized in Table S2. The elemental proportions on the surface of the LnMOFs are distinct from the bulk content acquired using SEM/EDS analysis, while the values of the carbon gained with the STEM are rather higher (carbonized sample).

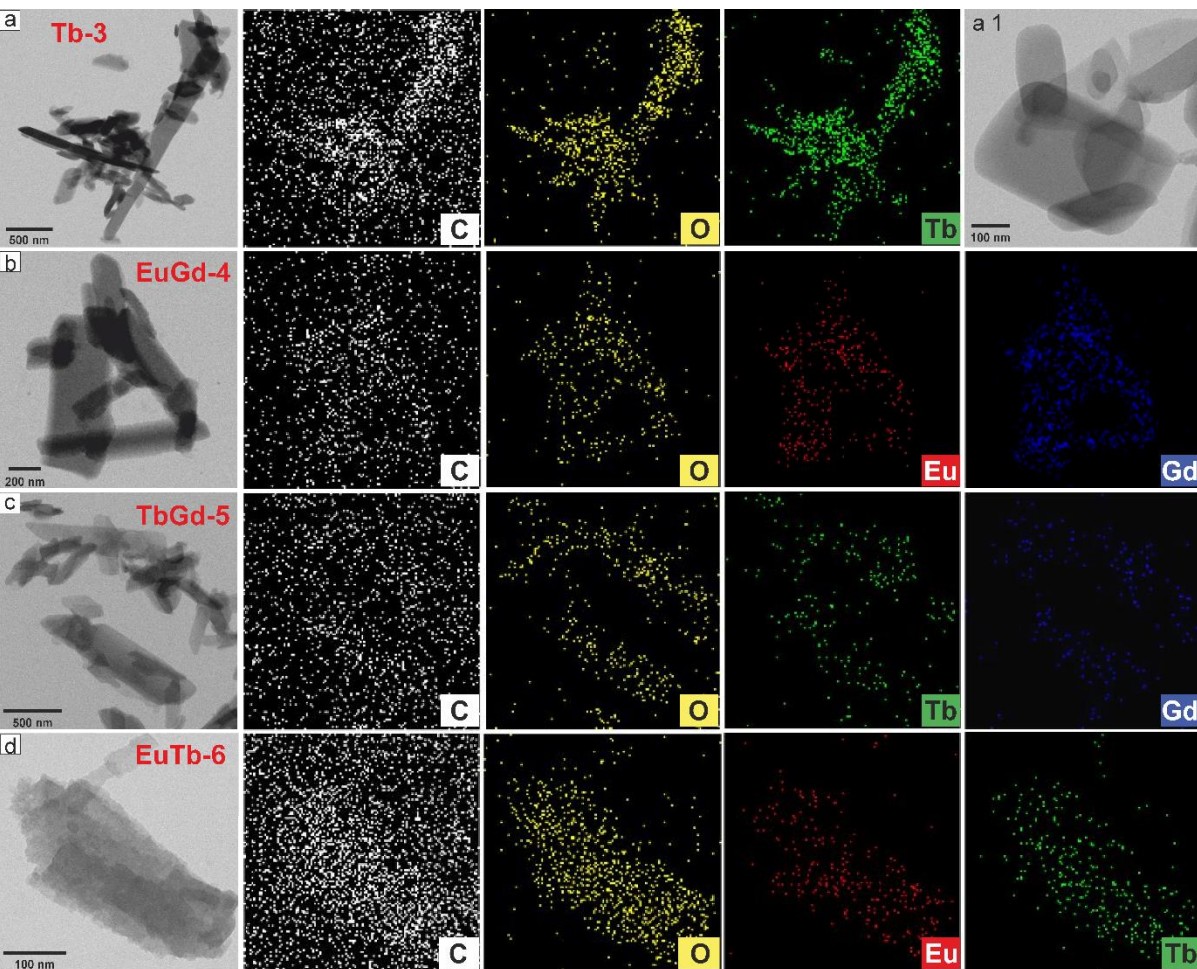

**Figure 6.** STEM images and EDS elemental mapping of the LnMOF powders: (**a**) Tb-3, (**b**) EuGd-4, (**c**) TbGd-5 and (**d**) EuTb-6 samples.

### 3.4. Luminescence of the LnMOFs

The luminescence tests were performed at room temperature. The excitation and emission spectra of the samples were recorded in a solid state. The excitation spectra of all of the LnMOFs provided a wide peak between 250 and 320 nm, which contained two top bands at 260 and 300 nm, cognated to the $Ln^{3+}$-$O^{2-}$ charge-transfer (CT) band and to the $\pi$–$\pi^*$ electron transformation of the organic ligand, respectively [25,38,39].

When monitored at 543 nm (Figure 7a), the EuTb-6 and TbGd-5 MOF samples presented terbium transitions from the $^7F_6$ ground to the $^5D_1$ (325 nm); $^5L_{7,8}$,$^5G_3$ (340 nm); $^5L_9$,$^5D_2$,$^5G_5$ (351 nm); $^5L_{10}$ (368 nm); and $^5G_6$,$^5D_3$ (376 nm) and $^5D_4$ (487 nm) [25,40], whereas when monitoring at 617nm (Figure 7b), EuTb-6 and EuGd-4 presented bands attributed to europium transitions from the $^7F_1$ state to the $^5D_1$ excited state at 535 nm, and from the $^7F_0$ ground state to the $^5D_1$ (524 nm); $^5D_2$ (464 nm); $^5D_3$ (417 nm); $^5L_6$ (392 nm); $^5G_2$ (383 nm); $^5D_4$ (361 nm) and $^5H_6$ (317 nm) [41–44]. However, for the EuTb-6 MOF sample, even monitoring the Eu3+ ion at 617 nm, $Tb^{3+}$ transitions were found from the $^7F_6$ ground to the $^5D_1$ (325 nm); $^5L_{7,8}$,$^5G_3$ (340 nm); $^5L_9$,$^5D_2$,$^5G_5$ (351 nm); $^5L_{10}$ (368 nm); $^5G_6$,$^5D_3$ (376 nm) and $^5D_4$ (487 nm), consequent from the energy between the $Tb^{3+}$ and $Eu^{3+}$ ions [25,38].

Figure 8a presents the emission spectra of the $Eu^{3+}$ and $Tb^{3+}$ ions of the mixed LnMOF samples. The spectra include the characteristic peaks of the $Eu^{3+}$ ion for the EuTb-6 and EuGd-4 samples, which were assigned to transitions from the $^5D_0$ excited state to the $^7F_J$ (J = 0, 1, 2, 3 and 4) ground states at 579, 591, 617, 653 and 702 nm, respectively. Although the $^5D_0 \rightarrow {}^7F_0$ transformation is exactly forbidden according to the standard Judd–Ofelt theory, its occurrence can be explicated using J-mixing due to the crystal-field disorder, and

its attendance shows that the $Eu^{3+}$ ion takes place with $C_{nv}$, $C_n$ or $C_s$ symmetry [44–46]. The symmetrical profile of this band indicates only one component, confirming that the emission of the $Eu^{3+}$ ion occurs from a single site in the matrix. Furthermore, as the band corresponding to the hypersensitive transition $^5D_0 \rightarrow {}^7F_2$ is more intense than the band due to the magnetic dipole transition $^5D_0 \rightarrow {}^7F_1$, the $Eu^{3+}$ occupies a site without an inversion center [44,46]. For the EuTb-6 and TbGd-5 samples, bands related to the $Tb^{3+}$ ion were observed at 492, 543, 585, 620 and 656 nm, and they were ascribed to the $^5D_4 \rightarrow {}^7F_J$ (J = 6, 5, 4, 3 and 2) transitions, respectively [25,40].

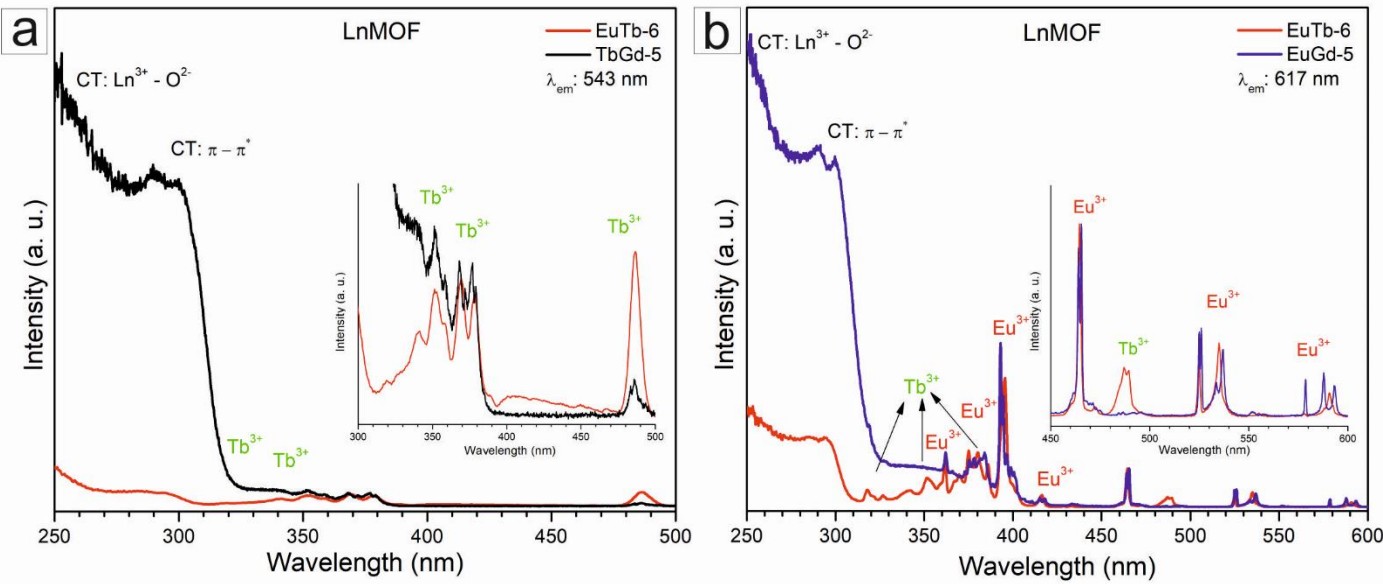

**Figure 7.** Excitation spectra of the LnMOF sample:s (**a**) EuTb-6 and TbGd-5 ($\lambda_{em}$ = 543 nm), and (**b**) EuTb-6 and EuGd-4 ($\lambda_{em}$ = 617 nm).

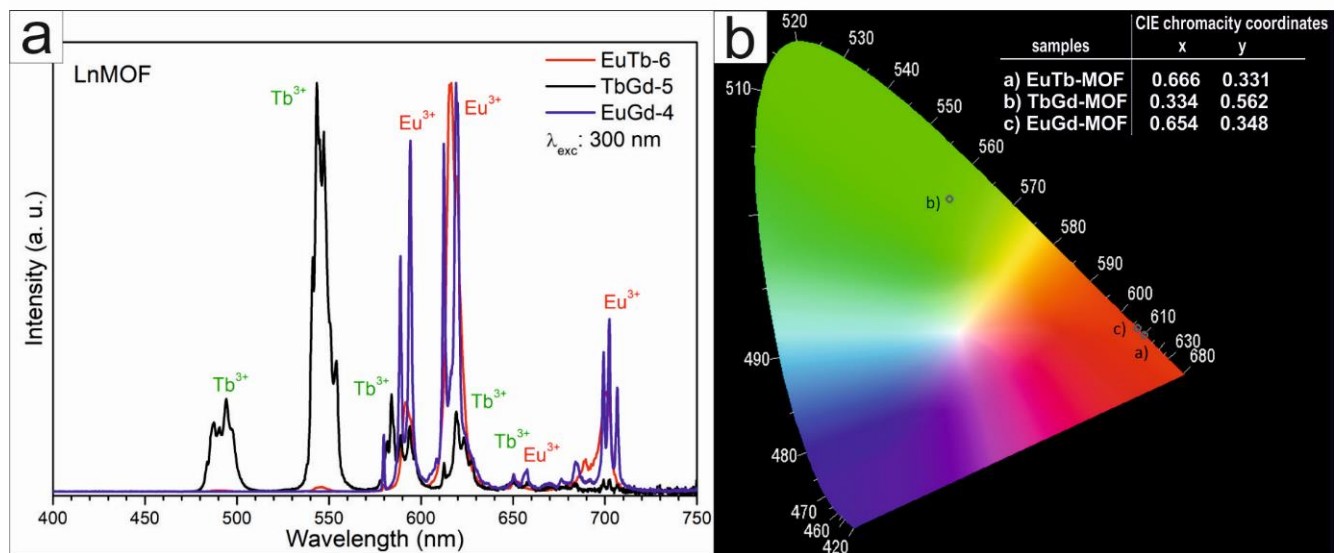

**Figure 8.** (**a**) Emission spectra and (**b**) CIE chromaticity diagrams of bimetallic lanthanide MOFs.

The CIE chromaticity coordinates were created by Spectra Lux 2.0 software [47] and the corresponding emission spectra acquired at room temperature (Figure 8b). As anticipated, the EuTb-6 and EuGd-4 samples showed CIE chromaticity coordinates closer to the default NTSC values (x = 0.670 and y = 0.330) [48], but were better than the values desired for use in commercial phosphors than $Y_2O_2S:Eu^{3+}$, which presents x = 0.64 and y = 0.35 [49]. The values of the CIE coordinates were x = 0.666 and y = 0.331, and x = 0.654 and y = 0.348 for

the EuTb-6 and EuGd-4 samples, respectively. For the TbGd-5 sample, the CIE chromaticity coordinates were x = 0.334 and y = 0.562, presenting emissions in the green region of the chromaticity diagram. The applications of the LnMOFs are listed in Table S3. The bimetallic LnMOFs can be used for applications as light-emitting materials [10,38], thermal sensors [20] or prototypical sensor to determine the concentrations of mixed organic compounds [17,21,22,24] and $Fe^{3+}$ ions [23].

## 4. Conclusions

In summary, the isostructural lanthanide metal–organic frameworks (LnMOFs (Ln = Eu, Gd, Tb, $Eu_{0.5}Gd_{0.5}$, $Gd_{0.5}Tb_{0.5}$ and $Eu_{0.5}Tb_{0.5}$)) were prepared by the solvothermal synthesis route. The series of nanostructured [Ln(btc)] (BTC: 1,3,5-benzenetricarboxylate) were obtained using sodium acetate as a modulator. XPS corroborated the coordination impact among the europium, gadolinium and terbium ions and the BTC ligand. XPS established the presence of two valence—Eu and Tb—states ($Eu^{3+}/Eu^{2+}$) and ($Tb^{3+}/Tb^{4+}$) and a single valence, Gd, state ($Gd^{3+}$). The various SEM and TEM morphologies of bimetallic [Ln(btc)] ($Eu_{0.5}Gd_{0.5}$, $Gd_{0.5}Tb_{0.5}$ and $Eu_{0.5}Tb_{0.5}$) were changed depending on the synthesis chemistry composition of $Ln^{3+}$ ions from nanorods (100–300 nm) for [$Eu_{0.5}Gd_{0.5}$(btc)] and [$Gd_{0.5}Tb_{0.5}$ (btc)] to nanoparticles (50–100 nm) for [$Eu_{0.5}Tb_{0.5}$(btc)]. The characteristic transitions within the 4f shells of the $Ln^{3+}$ ions were shown in the luminescence spectra. The MOFs (EuTb and EuGd) presented CIE chromaticity coordinates in the red region. This CIE result (x = 0.666 and y = 0.331) for the EuTb-MOF was expected due to its emissions from the europium ion, which were more intense than the terbium ion emissions. For TbGd-MOF, the CIE coordinates showed emissions in the green region of the chromaticity diagram. The mixed LnMOFs were prepared in order to extend their potential application in bifunctional luminescent sensors.

**Supplementary Materials:** The following are available online at https://www.mdpi.com/article/10.3390/inorganics9100077/s1. Figure S1: FTIR spectra of LnMOFs prepared by solvothermal synthesis. Figure S2: XRD patterns of LnMOFs prepared by solvothermal synthesis. Figure S3: XPS survey spectra of LnMOFs prepared by solvothermal synthesis. Table S1: XPS elemental atomic % of the LnMOF samples. Figure S4: SEM morphology and TEM images (in insert) of TbMOF powders prepared by solvothermal synthesis (a) (DMF/H2O) and (c) (DMF/H2O/NaOAc), and (b,d) EDS spectra of TbBTC. Table S2: EDS analysis of Tb-3, EuGd-4, TbGd-5 and EuTb-6 samples from the TEM/EDS spectra.

**Author Contributions:** Investigation, conceptualization, resources, writing—original draft preparation, H.B.; writing—review and editing, methodology, H.B., E.M. and L.R.; investigation and formal analysis, E.N., W.N., H.K., M.L., A.K., Z.M., M.S. and L.M. All authors have read and agreed to the published version of the manuscript.

**Funding:** The research was supported by the Grant Agency of the Slovak Academy of Sciences through project VEGA No. 2/0037/20 and APVV-20-0299. The authors thank Coordenação de Aperfeiçoamento de Pessoal de Nível Superior-Brasil (CAPES)-Finance Code 001, Conselho Nacional de Desenvolvimento Científico e Tecnológico (CNPq, grants: 302702/2018-0 L.A.R. and 302668/2017-9 E.J.N.).

**Data Availability Statement:** The data presented in this study are available on request from the corresponding author. This data is not publicly available due to its excessive size and complex format.

**Conflicts of Interest:** The authors declare no conflict of interest.

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
