# Peer review of "Nanostructure and Luminescent Properties of Bimetallic Lanthanide Eu/Gd, Tb/Gd and Eu/Tb Coordination Polymers"

_inorganics, doi:10.3390/inorganics9100077_

Round 1
Reviewer 1 Report
The article entitled “Nanostructure of Bimetallic Lanthanide Eu/Gd, Tb/Gd and Eu/Tb Metal-Organic Frameworks as Luminescent Sensors” by Brunckova et al., intends to present the synthesis and luminescence investigations for three “MOFs type” compounds with lanthanide (Eu,Gd,Tb) and other three “MOFs type” bimetallic compounds (Eu/Gd, Tb,Gd, Eu/Tb) with BTC used as ligand.
First of all, I disagree with the authors' assertion that the compounds provided in this manuscript are metal organic frameworks (MOFs). Even if the authors intended to characterize these lanthanide compounds by using XPS and XRD analysis, the data presented into the manuscript are inconsistent and has not enough accuracy.
The study requires more detailed investigations to propose these lanthanides compounds as metal-organic frameworks. The authors present the XRD powder of these compounds as similar with LnMOF, beeing isostructural compounds, see page 127, without a X-ray single crystal measurement which is of most importance to characterize MOFs structures. The XPS study is known as a technique for analyzing a material's surface and doesn't prove that the structure of these compounds consists of a regular array of lanthanide ions surrounded by a BTC linker. Even the XRD study is not correctly correlated with the literature.
Major questions on the paper:
Line 2: The title is inadequate with the content of the manuscript. Again I should point out that not all mixtures with lanthanides and ligands lead to MOFs products and this is the case for this study too.
Line 15: It is only general information about MOFs; should be in the introduction. . Please reformulate to provide accurate information regarding the identified products.
Line 21-22: Please explain how the” nanostructure of mixed LnMOFs” could be related to luminescence propriety.
Line 35: the reference [3] is inappropriate.
Line 52: Has been established already in line 19 that LnMOFs are noticed as LnBTC; please avoid repetition.
Line 53: Please explain why BTC is classical??
Line 65: replace “Eu / GdMOF” with “Eu/GdMOF”;
Paragraph 96-109: There are many concerns about the preparation of these “MOFs”: the amount of reactants, the ratio between Ln:TBC not presented, isolation process, purification process, how/if was separate from the un-reacted reactants, the yields; elemental analysis, reproducibility. The preparation of these compounds is too important to be just simply presented as it is in this paper. The amount of reactants should be added, the isolation procedure should be described, the purification procedure should be presented, and a comparison with other methods and procedures already reported in the literature should be introduced.
Also please point out if there have been performed several experiments for each compound (1-7) and provide more information regarding the reproducibility of the results of the produced samples. Reproducibility is essential for MOFs. Also please provide some information about their stability.
How authors could explain why the conversion of the ligand is total ?? The unreacted compounds could be present in the synthesized compounds as Tb0.5Gd0.5-MoF or Eu0.5Tb0.5 MOF??
Line 117: Please add the name of the EDX instrument.
Paragraph 126-135: How can you explain the peaks at 1750 -1680 cm-1 from the Figure S1 (Supplementary information)?? The FTIR spectra are not explained in detail and the presence of some peaks could suggest the presence of BTC ligand, and to my opinion, the purity of the products was not proved.
Paragraph 136-146: Here is the main concern regarding the purity and composition of these products. The authors cite their previous paper [20] indeed, but the data presented here are identical to that reported in ref. 20. It cannot be accepted this sentence ''almost identical peaks, which confirm their coincident crystal structure" reference [12,23], because in the figure S2 are a lot of missing data as follow: the 5-10 (2theta) interval; simulated patterns; and also the fingerprint of XPD patterns do not coincide with those from the cited reference[12,23].
The XPD patterns presented in ref[12, 23, 33] or simulated XPD from X-ray single crystal of Eu(BTC) - (CCDC 290771) are not fitting with XPD patterns presented in this manuscript and they must fit. This is a strong doubt that the structure of these products is not as presented in the article. The simulation patterns would prove the similarity of the signals, thus must be included in the paper.
Figure 2: Due to the fact that EuGd-4 covers the signals of compounds 5 and 6, please reduce the size of EuGd-4 signal.
Line 140 The XRD patterns of Gd-2 is not ‘ín good agreement’ with ref [5] as was suggested, please change the reference.
Line 141: there is a mix of the results with the typical introduction text. Please rewrite it; the sentence is not correlated with the data reported here.
Line 142-146: There are only reported data that are not correlated with the new data (for introduction), again is mixing results with the introduction typical text. Please rewrite.
Paragraph 149-176: Why it was decided to do the peak fitting XPS high resolution spectra of Eu 4d, Tb 4d and not 3d orbital?? Why the 3d orbitals presented in figure 1 are not deconvoluted?
Eu+2 could be a surface defect?? Please explain clear the presence of both valences.
Figure 1 – fitting (deconvolution) of the high-resolution spectra for each element;
Figure 2: move the figure in SI
Figure 3: In Figure 3 for C 1s are presented bonds C=O; in O 1s Eu-C-O and N 1s O-N-C-H whose presence is not justified; please explain their presence.
Paragraph 203-214: If the rods of TbMOF are in the range 2-10µm that means 2000-100000 nm and that is not in concordance with Table 1. How do you explain that??
Figure 4 b presents the element At% for TbGd-5 that are not in concordance with table S2. The EDS proves for probe 4, show the composition for TbGd-5, 3.3 at 3.2, but in the Table S2 seems to be different (3.0 at 2.2). Please explain the discrepancies.
Also, the graphical representation released in Figure 4 (b and d) have data different than those presented in Table S2.
In the figures 4 and S4, the nitrogen is present. Please explain the presence of nitrogen. It could be from nitrate or DMF?
Line 226: the sentence “The records acquired using TEM were fully alike and well agreed with the results gained from SEM” is not really sustained. In Figure 1 are presented TgGd-5 with rods at ~10-20 µm, and in SEM are 200 nm. As previously mentioned (line 203) the SEM and TEM are surface morphology methods and can not elucidate the “homogeneous partition” as was written line 245. Please reconsider the paragraph.
Figure 5: figure 5 d is not clear. Please explain in detail that figure?
Line 245: Please be consistent with STEM or TEM meaning. Please use the same terminology or describe the instrument in the instrumental part.
Paragraph 256-298: Why was included in the manuscript the luminescence investigations only from probes 5,6, and 7, and not also from Eu-1 and Tb -3,? The excitation and emission spectra were recorded in solid-state? Please mention that. Have you recorded data also for BTC? Please introduce in the manuscript.
Line 276: The characteristic peaks recorded in the luminescence spectra can be from the Ln+3 and from lanthanide nitrate. Please explain why you excluded the signal from lanthanide nitrate.
Line 298: Please move this sentence to the conclusion part.
Table 2 can be moved to SI content.
Could you please add the thermogravimetric Analysis (TG/DTG) as you presented in the previous paper from the group [ref 20]
The manuscript needs more and careful checks of the characterization of the product.
Author Response
The Responses to Reviewers (Manuscript Inorganics-1391218)
Title: Nanostructure and Luminescent Properties of Bimetallic 2 Lanthanide Eu/Gd, Tb/Gd and Eu/Tb Coordination Polymers
Dear Reviewers,
Thank you for your comments for the manuscript Inorganics-1391218 by H. Brunckova et al. I have made some revisions by following your valuable suggestions.
Reviewer #1:
Comments and Suggestions for Authors
The article entitled “Nanostructure of Bimetallic Lanthanide Eu/Gd, Tb/Gd and Eu/Tb Metal-Organic Frameworks as Luminescent Sensors” by Brunckova et al., intends to present the synthesis and luminescence investigations for three “MOFs type” compounds with lanthanide (Eu,Gd,Tb) and other three “MOFs type” bimetallic compounds (Eu/Gd, Tb,Gd, Eu/Tb) with BTC used as ligand.
First of all, I disagree with the authors' assertion that the compounds provided in this manuscript are metal organic frameworks (MOFs). Even if the authors intended to characterize these lanthanide compounds by using XPS and XRD analysis, the data presented into the manuscript are inconsistent and has not enough accuracy.
The study requires more detailed investigations to propose these lanthanides compounds as metal-organic frameworks. The authors present the XRD powder of these compounds as similar with LnMOF, beeing isostructural compounds, see page 127, without a X-ray single crystal measurement which is of most importance to characterize MOFs structures. The XPS study is known as a technique for analyzing a material's surface and doesn't prove that the structure of these compounds consists of a regular array of lanthanide ions surrounded by a BTC linker. Even the XRD study is not correctly correlated with the literature.
A: Thank you for comments and suggestions. The Introduction, results and conclusions were improved. These changes were incorporated in the Manuscript highlighted in yelow color.
Major questions on the paper:
Line 2: The title is inadequate with the content of the manuscript. Again I should point out that not all mixtures with lanthanides and ligands lead to MOFs products and this is the case for this study too.
A:The title was changed to Nanostructure and Luminescent Properties of Bimetallic Lanthanide Eu/Gd, Tb/Gd and Eu/Tb Coordination Polymers.
Line 15: It is only general information about MOFs; should be in the introduction. Please reformulate to provide accurate information regarding the identified products.
A: The term coordination network solids can be seen as a compromise: IUPAC nomenclature can be adhered to even if coordination polymer is avoided. Metal-organic frameworks will thus be a subclass of coordination network solids, which in its turn is a subclass of coordination polymer. An alternative classification that avoids introducing new terms is to adopt a very broad inclusive definition of metal-organic framework as: “any system that forms a 2D or 3D network with carbon containing ligands bridging mononuclear, polynuclear or 1D coordination entities” [Ref x]. https://doi.org/10.1039/C2CE06488J
The sentence was reformulated. The study presents synthesis, structural and luminescence properties for lanthanide metal-organic frameworks (LnMOFs), which belong to a sub-class of coordination polymers.
Line 21-22: Please explain how the” nanostructure of mixed LnMOFs” could be related to luminescence propriety.
A: Luminescent nanomaterials, specifically assembled nano-architectures, demonstrate a better performance in terms of photoluminescence response together with customized microstructures and morphologies compared to bulk counterparts [Ref 2]. https://doi.org/10.3390/ma14164591 The properties and applications of luminescent materials are strongly dependent on their chemical composition, crystal structure, size and morphology. The Zn2GeO4:Mn2+ nanobundles exhibit broader diameters and larger particle size, which induces lower surface area and leads to fewer defects and stronger luminescence [Ref 3]. https://doi.org/10.1039/C8RA08636B
Line 35: the reference [3] is inappropriate.
A: The reference was changed.
Line 52: Has been established already in line 19 that LnMOFs are noticed as LnBTC; please avoid repetition.
A: Thank you for suggestion.The sentence was corrected.
Line 53: Please explain why BTC is classical??
A: Zhang et al. used term classical BTC in literature [Ref 4]. https://doi.org/10.1039/x0xx00000x BTC ligand is classical because it is used in various coordinating polymers with metallic Cu, Pb, Ni, Fe ions in addition to Ln ions. The solubility of H3BTC acid in pure water, isopropyl alcohol, isobutyl alcohol, methanol, ethanol, and ethylene glycol is good at the temperature range from 298 to 360 K.
Line 65: replace “Eu / GdMOF” with “Eu/GdMOF”;
A: The word was corrected.
Paragraph 96-109: There are many concerns about the preparation of these “MOFs”: the amount of reactants, the ratio between Ln:TBC not presented, isolation process, purification process, how/if was separate from the un-reacted reactants, the yields; elemental analysis, reproducibility. The preparation of these compounds is too important to be just simply presented as it is in this paper. The amount of reactants should be added, the isolation procedure should be described, the purification procedure should be presented, and a comparison with other methods and procedures already reported in the literature should be introduced.
A: The paragraph was corrected. The amount of reactants, the ratio between Ln:TBC, isolation process, purification process, the yields, reproducibility and comparison with other methods and procedures were supplemented.
LnMOFs were prepared via modified solvothermal synthesis [3,11,12] pursuant to the previous work [20]. Lanthanide(III) nitrate hydrate Ln(NO3)3·xH2O (1.0 mmol) and H3BTC (0.21 g, 1.0 mmol) were dissolved in the 30 mL mixture of DMF/H2O (1:1v/v) solvents (Ln = Eu, Gd, Tb) together with the modulator NaOAc (0.3 mmol). The preparation procedures for Eu. Gd and Tb lanthanide MOFs were same and were performed using different starting nitrates using Eu(NO3)3. 5H2O (0.443 g), Gd(NO3)3. 6H2O (0.448 g) and Tb(NO3)3. 5H2O (0.449 g). The ratio between Ln:TBC is 0.36. The three solutions of EuBTC (Eu-1), GdBTC (Gd-2) and TbBTC (Tb-3) were mixed at 25°C for 1 h and heated at 60°C for 48 h and then cooled to room temperature to give white (Eu-1 and Gd-2) and colorless (Tb-3) crystals. After synthesis, the products were isolated by centrifugation and washed several times with ethanol and water, respectively and then dried in air. The prepared Eu-1, Gd-2 and Tb-3 resulted with yield of 66% (0.319 g), 69% (0.270 g) and 70% (0.339 g), respectively without elemental analysis. The preparation of mixed bimetallic Eu0.5Gd0.5BTC (EuGd-4), Tb0.5Gd0.5BTC (TbGd-5) and Eu0.5Tb0.5BTC (EuTb-6) is the same to that for simple LnBTC, only pure Ln nitrate was exchanged via mixture of two corresponding nitrates. For preparation of EuGd-4 powder, the nitrate of Eu (0.223 g, 0.5 mmol) and Gd (0.224 g, 0.5 mmol) with BTC (0.21 g, 1.0 mmol) and NaOAc (0.03 g, 0.3 mmol) were dissolved in 30 mL DMF/H2O. For TbGd-5 and EuTb-6 syntheses was used Tb(NO3)3·6H2O (0.225 g, 0.5 mmol). The bimetallic LnMOF powders were formed after heating at 60°C for 48 h. The preparation procedures for other lanthanide MOFs were analogous. The experimental synthesis of each samples was repeated for three times. The syntheses of Tb-3 and TbGd-5 MOFs were performed without the NaOAc modulator for SEM and TEM analysis.
The comparison with other methods and procedures for solvothermal synthesis of LnBTC (Ln= Eu0.5/Gd0.5 or Tb0.5/Gd0.5) were used the selected lanthanide chloride salts, sodium trifluoroacetate (NaTFA), and BTC in ~1:0.9:0.6 Ln:TFA:BTC ratio with the solvents water and DMF [7] and ball milling preparation of Eu0.5/Gd0.5(BTC) or Tb0.5/Gd0.5(BTC) with the H3BTC and respective lanthanide carbonate hydrates Ln2(CO3)3·xH2O in 2:1 molar ratio. The other solvothermal synthesis of LnMOFs have been reported with various ligands as psa [15], H2OBA [16], acid H2FDA [17], H4L+Cl- [18] and NDC [19].
Also please point out if there have been performed several experiments for each compound (1-7) and provide more information regarding the reproducibility of the results of the produced samples. Reproducibility is essential for MOFs. Also please provide some information about their stability.
A: The experimental synthesis for each compound (1-6) was repeated for three times. The prepared Eu-1, Gd-2 and Tb-3 resulted with yield of 66% (0.319 g), 69% (0.270 g) and 70% (0.339 g), respectively without elemental analysis. EuGd-4, TbGd-5 and EuTb-6 resulted with yield of 65%. All the samples had similar thermal stability up to 540-600°C. The Figure S1 DSC/TG curves of LnMOFs (LnBTC) prepared by solvothermal synthesis was included in the manuscript.
How authors could explain why the conversion of the ligand is total ?? The unreacted compounds could be present in the synthesized compounds as Tb0.5Gd0.5-MoF or Eu0.5Tb0.5 MOF??
A: The solutions of LnBTC were mixed at 25°C for 1 h and heated at 60°C for 48 h, until complete dissolution. The products were isolated by centrifugation and washed several times with ethanol and water, respectively and then dried in air. We assume that the unreacted compounds could be present in the synthesized compounds as Tb0.5Gd0.5-MoF or Eu0.5Tb0.5 MOF in small quantities. Some unreacted residues of MOF compounds that were not involved in the polymerization may be dispersed in the polymer matrix.
Line 117: Please add the name of the EDX instrument.
A: Surface morphologies were characterized using scanning electron microscopy (SEM), (Auriga Compact, Carl Zeiss Germany) and high resolution transmission electron microscopy (TEM), (JEOL-JEM 2100F) and energy dispersive X-ray (EDS, Oxford Instruments X-max80 SDD detector) spectroscopy.
Paragraph 126-135: How can you explain the peaks at 1750 -1680 cm-1 from the Figure S1 (Supplementary information)?? The FTIR spectra are not explained in detail and the presence of some peaks could suggest the presence of BTC ligand, and to my opinion, the purity of the products was not proved.
A: The peaks at 1750 -1680 cm-1can be explain: In spectra, the characteristic bands of carboxyl group of BTC disappear (3090 and 1720 cm-1) and new peaks appear at 1568-1538 and 1386, cm-1, which belong to the stretching nas(COO-) and ns(COO-) vibrations of the carboxylic ions.
In spectrum of H3BTC acid, the characteristic bands of carboxyl group of BTC are at 3090, 1720, and 537 cm-1. In the spectra of MOFs, no band at 1720 cm-1 corresponding to the COOH groups is seen, designating the complete deprotonation of the carboxylic acid and coordination of COO- groups to the lanthanide centre. Peak at 1678 cm-1 belongs to ν (C=O) of DMF.
Paragraph 136-146: Here is the main concern regarding the purity and composition of these products. The authors cite their previous paper [20] indeed, but the data presented here are identical to that reported in ref. 20. It cannot be accepted this sentence ''almost identical peaks, which confirm their coincident crystal structure" reference [12,23], because in the figure S2 are a lot of missing data as follow: the 5-10 (2theta) interval; simulated patterns; and also the fingerprint of XPD patterns do not coincide with those from the cited reference[12,23].
The XPD patterns presented in ref[12, 23, 33] or simulated XPD from X-ray single crystal of Eu(BTC) - (CCDC 290771) are not fitting with XPD patterns presented in this manuscript and they must fit. This is a strong doubt that the structure of these products is not as presented in the article. The simulation patterns would prove the similarity of the signals, thus must be included in the paper.
A: The text was corrected. The simulation patterns were not performed.
Figure S2: Due to the fact that EuGd-4 covers the signals of compounds 5 and 6, please reduce the size of EuGd-4 signal.
A: The Figure S2 was corrected.
Line 140 The XRD patterns of Gd-2 is not ‘ín good agreement’ with ref [5] as was suggested, please change the reference.
A: The Ref [5] was changed to Ref [11].
Line 141: there is a mix of the results with the typical introduction text. Please rewrite it; the sentence is not correlated with the data reported here.
A: The sentence was deleted.
Line 142-146: There are only reported data that are not correlated with the new data (for introduction), again is mixing results with the introduction typical text. Please rewrite.
A: The text was rewrited.
The XRD diffractograms of samples are shown in Figure S2. The XRD peaks are in good compliance with previous report indicating that obtained MOFs are isostructural [20]. XRD diffractograms are in good agreement with patterns of Ln(BTC), which have tetragonal structure [11]. The Eu(BTC)(H2O)DMF, built with Eu3+ cation and the organic linker BTC via coordination bonding, is 3D framework with tetragonal structure [3,4,11]. XRD shows that Eu(BTC)(H2O)DMF structure is not changed using slight addition of NaOAc, but ever it will be transformed to monoclinic structure in the effect of major quantity of modulator agent [2,3].
Paragraph 149-176: Why it was decided to do the peak fitting XPS high resolution spectra of Eu 4d, Tb 4d and not 3d orbital?? Why the 3d orbitals presented in figure 1 are not deconvoluted?
A: I decided to presenting XPS high resolution spectra of Eu 4d, Tb 4d, because in available references are infrequent. The 3d orbitals presented in Figure 1 are not deconvoluted, because in the previous work [20] they are mentioned.
Eu+2 could be a surface defect?? Please explain clear the presence of both valences.
A: I think, that the Eu+2 is not surface defect. The high resolution Gd, Eu and Tb 3d was measured at HV 20 and HV 50 (mean resolution of the machine).
The presence of two valence states Eu3+/Eu2+ and Tb3+/Tb4+ in the corresponding samples was confirmed. It can also be caused by preparation of samples and drying in air.
Figure 1 – fitting (deconvolution) of the high-resolution spectra for each element;
A: The deconvolution of C 1s, N 1s and O 1s peaks was visible in Figure 3. Fitting of the high-resolution spectra of Eu, Gd aTb are in previous worwork [20].
Figure 2: move the figure in SI
A: The Figure 2 was inluded to SI as Figure S4.
Figure 3: In Figure 3 for C 1s are presented bonds C=O; in O 1s Eu-C-O and N 1s O-N-C-H whose presence is not justified; please explain their presence.
A: In Figure 3 for C 1s are presented bonds C=O; in O 1s, Eu-C-O; and N 1s O-N-C-H; caused by the presence of DMF and BTC.
The XPS results for C 1s and N 1s are in good agreement with the FTIR n(C=O) and d (O=C-N) from DMF coordinated, respectively.
Paragraph 203-214: If the rods of TbMOF are in the range 2-10µm that means 2000-100000 nm and that is not in concordance with Table 1. How do you explain that??
A: In Figure S4, the TbMOF morphology without and with modulator NaOAc is presented.
The TbMOF morphology is formed typical rods in the range 2 to 10 mm without modulator (Figure S4a [10]. In Figure S4b, needle-shaped nanocrystals of TbMOF after addition of modulator NaOAc are observed [13].
Figure 4 b presents the element At% for TbGd-5 that are not in concordance with table S2. The EDS proves for probe 4, show the composition for TbGd-5, 3.3 at 3.2, but in the Table S2 seems to be different (3.0 at 2.2). Please explain the discrepancies.
A: Figure 4a,b presents SEM/EDS ( C, O, N and Ln) morphology of TbGd-5 prepared by solvothermal synthesis without modulator NaOAc. In Table S2 is EDS analysis of TbGd-5 sample from STEM/EDS spectra of sample prepared using modulator. The elemental proportions on the surface of LnMOFs distinct from bulk content acquired using SEM/EDS analysis (C, O, Ln), while the values of carbon gained with the STEM rather higher (carbonized sample).
Also, the graphical representation released in Figure 4 (b and d) have data different than those presented in Table S2.
A: In Figure 4c,d (SEM/EDS) are data different to in Table S2 (STEM/EDS).
In the figures 4 and S4, the nitrogen is present. Please explain the presence of nitrogen. It could be from nitrate or DMF?
A: The presence of nitrogen is from DMF. I assume that nitrates have reacted with BTC.
Line 226: the sentence “The records acquired using TEM were fully alike and well agreed with the results gained from SEM” is not really sustained. In Figure 1 are presented TgGd-5 with rods at ~10-20 µm, and in SEM are 200 nm. As previously mentioned (line 203) the SEM and TEM are surface morphology methods and can not elucidate the “homogeneous partition” as was written line 245. Please reconsider the paragraph.
A: The Figure 4a,b presented TbGd-5 with rods at ~10-20 µm (prepared without modulator) and in Table 1 are the variation in sizes of nanorods obtained from the TEM images (Figure 5).
The records acquired using TEM were fully alike and well agreed with the results of samples prepared solvothermal synthesis with addition of modulator (NaOAc) gained from SEM.
The STEM and EDS mapping of elements in LnMOFs (Figure 6) elucidate the homogeneous partition of Ln, O and C elements through the surface of samples [32].
Figure 5: figure 5 d is not clear. Please explain in detail that figure?
A: The Figure present one rod which small nanorods (length 300-450 nm and width 50-100 nm) are incorporated.
In Figure 5d present one rod with length 850 nm and width 400 nm, which small nanorods (length 300-450 nm and width 50-100 nm) are incorporated.
Line 245: Please be consistent with STEM or TEM meaning. Please use the same terminology or describe the instrument in the instrumental part.
A: Scanning transmission electron microscope (STEM) is type of transmission electron microscope (TEM). I include Scanning transmission electron microscope (STEM) as type of transmission electron microscope (TEM).
Surface morphologies were characterized using scanning electron microscopy (SEM), (Auriga Compact, Carl Zeiss Germany) and high resolution transmission electron microscopy (TEM), (JEOL-JEM 2100F), Scanning transmission electron microscope (STEM) and energy dispersive X-ray (EDS, Oxford Energy TEM 250) spectroscopy.
Paragraph 256-298: Why was included in the manuscript the luminescence investigations only from probes 5,6, and 7, and not also from Eu-1 and Tb -3,? The excitation and emission spectra were recorded in solid-state? Please mention that. Have you recorded data also for BTC? Please introduce in the manuscript.
A: Luminescence investigations were not performed in samples Eu-1, Gd-2 and Tb-3. We have not recorded data for BTC.
The excitation and emission spectra of samples were recorded in solid-state.
Line 276: The characteristic peaks recorded in the luminescence spectra can be from the Ln+3 and from lanthanide nitrate. Please explain why you excluded the signal from lanthanide nitrate.
A: The lanthanide nitrate were not determined in LnMOFs, prepared by solvothermal synthesis using mixture of solvents EtOH/H2O [20]. XPS and SEM/EDS investigations confirmed no presence of nitrogen. Our samples were prepared by a similar procedure with the same amount of reactants with change in DMF/H2O solvents and the addition of a Modulator. We assume the absence of nitrates, which should react with H3BTC.
Line 298: Please move this sentence to the conclusion part.
A: The sentence was moved to the Conclusion part.
Table 2 can be moved to SI content.
A: Table 2 was moved to SI as Table S3.
Could you please add the thermogravimetric Analysis (TG/DTG) as you presented in the previous paper from the group [ref 20]
A: The thermogravimetric Analysis ( DSC/TG) was added as Figure 1.
Figure 1 DSC/TG curves and of LnMOF (Ln = Eu, Gd, Tb) prepared by solvothermal synthesis.
TG and DSC curves of LnMOFs (Eu-1, Gd-2, Tb-3) prepared by sovothermal synthesis using modulator (NaOc) are shown in Figure 1. The TG curves are similar, all of which display a two-step or three-step weight loss [3]. The initial weight loss starting at around 100°C up to 160°C observed for all samples can be ascribed to the loss of coordinated solvent molecules (DMF and H2O) [20]. From TG curves of Eu-1and Gd-2 shows two main steps of gradual weight loss process before 220°C, attributing to the release of H2O and two DMF molecules and 7.0 % in 220-410°C temperature range, corresponding to the loss of coordinated DMF molecules. Above 410°C, the complete framework sequentially begins to collapse upon further heating. All the samples had similar thermal stability up to 540-600°C. The weight loss at higher temperatures of 540-600°C is attributed to the thermal decomposition of LnBTC to Ln oxides. From DSC curves result that the end point of endo peak depends on chemical component of lanthanides (Eu-1: 620°C, Gd-2: 658°C and Tb-3: 673°C).
The manuscript needs more and careful checks of the characterization of the product.
A: Thank you for comments and suggestions. I have made careful checks and corrections.
The Figures and Tables in Response to Reviewer comments are marked as in the reviewer comments of uncorrected Manuscript. The references I revised Manuscript were corrected.
Figure Caption: in revised Manuscript Inorganics-1391218)
Figure 1 DSC/TG curves of LnMOF (Ln = Eu, Gd, Tb) prepared by solvothermal synthesis.
Figure 2 HR XPS spectra of Eu 3d, Gd 3d, Tb 3d, C 1s, O 1s and N 1s of LnBTC (Eu-1, Gd-2, Tb-3, EuGd-4, TbGd-5 and EuTb-6) powders.
Figure 3 HR XPS spectra of C 1s, O 1s and N 1s and curve fitted peaks for EuBTC and Eu0.5Tb0.5BTC powders.
Figure 4 SEM morphology and TEM images (in insert) of LnMOF powders prepared by solvothermal synthesis a)TbGd-5 (DMF/H2O), c) EuTb-6 (DMF/H2ONaOAc) and EDS spectra of b) Tb0.5Gd0.5BTC and d) Eu0.5Tb0.5BTC.
Figure 5 TEM images of LnMOF powders a) Eu-1, b) Gd-2, c) Tb-3, d) EuGd-4, e) TbGd-5 and f) EuTb-6.
Table 1 The variation in sizes of nanorods obtained from the TEM images (Figure 5).
Figure 6 STEM images and EDS elemental mapping of LnMOF powders a) Tb-3, b) EuGd-4, c) TbGd-5 and d) EuTb-6 samples.
Figure 7 Excitation spectra of LnMOF samples a) EuTb-6, TbGd-5 (λem = 543 nm) and b) EuTb-6, EuGd-4 (λem = 617 nm).
Figure 8 a) Emission spectra and b) CIE chromaticity diagrams of bimetallic lanthanide MOFs excited at 300 nm marked a) EuTb-6, b) TbGd-5 and c) EuGd-4.
Figure Caption in SI: in revised Manuscript Inorganics-1391218)
Figure S1. FTIR spectra of LnMOFs prepared by solvothermal synthesis.
Figure S2. XRD patterns of LnMOFs prepared by solvothermal synthesis.
Figure S3. XPS survey spectra of LnMOFs prepared by solvothermal synthesis.
Table S1. XPS elemental atomic % of LnMOF samples.
Figure S4 HR XPS spectra of Eu 4d and Tb 4d of EuBTC, TbBTC and Eu0.5Tb0.5BTC samples.
Figure S5. SEM morphology and TEM images (in insert) of TbMOF powders prepared by solvothermal synthesis a) (DMF/H2O), c) (DMF/H2O/NaOAc) and b,d) EDS spectra of TbBTC.
Table S2 EDS analysis of Tb-3, EuGd-4, TbGd-5 and EuTb-6 samples from TEM/EDS spectra.
Table S3 The applications of luminescent bimetallic LnMOFs.
Yours sincerely,
Helena Brunckova
Institute of Materials Research Slovak Academy of Sciences
Watsonova 47, 040 01 Kosice, Slovakia
E mail: hbrunckova@saske.sk
Reviewer #2:
A: Thank you for comments and suggestions. The Introduction, results and conclusions were improved. These changes were incorporated in the Manuscript highlighted in green color. English was corrected.
Comments and Suggestions for Authors
The manuscript by Brunckova et al. reports the synthesis and luminescent properties of a series of isostructural homo- and heterobimetallic Eu, Gd, Tb metal-organic frameworks. The MOFs were prepared by the procedure previously reported by the same authors and were characterized by XPS, SEM and EDS methods.
I believe the manuscript fits the scope of the journal, however, some corrections are advisable:
- In the introduction part, the authors give some general information on MOFs and Ln-MOF in particular. However, they cite particular works ([1]-[8]), rather than more general review articles. I suggest including some references to recent reviews on MOFs and Ln-MOFs. The authors may find useful these two reviews: 10.1070/RCR5026 and 10.3390/ma13122699
A: The two reviews were included in the Manuscript.
Coordination polymers (CPs) are constructed of metal ions and bridging ligands that combine them into solid-state structures extending in one (1D), two (2D), or three dimensions (3D). Two- and three-dimensional CPs with potential voids are often designated to as metal-organic frameworks (MOFs) [1,2].
[1] Kuznetsova, A.; Matveevskaya, V.; Pavlov, D.; Yakunenkov, A.; Potapov, A. Coordination Polymers Based on Highly Emissive Ligands: Synthesis and Functional Properties. Materials 2020, 13, 2699-2766. https://doi.org/10.3390/ma13122699
[2] Kovalenko, K.A.; A S Potapov, A.S.; Fedin, V.P. Micro- and mesoporous metal-organic coordination polymers for separation of hydrocarbons, Rus. Chem. Rev. 2021, 90. https://doi.org/10.1070/RCR5026.
- In Table S1 elemental composition of Ln-MOFs are given, however, no values calculated for the assumed formula Ln(BTC) are given, which does not allow to quickly assess the chemical purity of the products.
A: The XPS study is known as technique for analyzing surface of compounds and does not prove that the structure of these LnBTC consists of a regular array of lanthanide ions surrounded by a BTC linker. The hydrogen element was not specified.
The values calculated for the assumed formula Ln(BTC)(DMF)2(H2O), {Ln[C6H3(CO2)3](C3H7NO)2(H2O), C15H19N2O9Ln are C7HxN2O7Eu, C11.6HxN2.8O9Gd, C12.4HxN3.5O9.4Tb, C10.8HxN1.4O7.6 Eu0.52Gd0.48, C14.1HxN2.6O10.6 Tb0.47Gd0.53 and C14.1HxN2.3O8.9 Eu0.47Tb0.53.
- In the title, “Luminescent sensors” are mentioned, however, no sensor properties were studied in the work presented. I suggest changing this title from “luminescent sensors” to “luminescent properties”
A: The title was changed.
Nanostructure and Luminescent Properties of Bimetallic Lanthanide Eu/Gd, Tb/Gd and Eu/Tb Coordination Polymers.
Yours sincerely,
Helena Brunckova
Institute of Materials Research Slovak Academy of Sciences
Watsonova 47, 040 01 Kosice, Slovakia
E mail: hbrunckova@saske.sk

Reviewer 2 Report
The manuscript by Brunckova et al. reports the synthesis and luminescent properties of a series of isostructural homo- and heterobimetallic Eu, Gd, Tb metal-organic frameworks. The MOFs were prepared by the procedure previously reported by the same authors and were characterized by XPS, SEM and EDS methods.
I believe the manuscript fits the scope of the journal, however, some corrections are advisable:
- In the introduction part, the authors give some general information on MOFs and Ln-MOF in particular. However, they cite particular works ([1]-[8]), rather than more general review articles. I suggest including some references to recent reviews on MOFs and Ln-MOFs. The authors may find useful these two reviews: 10.1070/RCR5026 and 10.3390/ma13122699
- In Table S1 elemental composition of Ln-MOFs are given, however, no values calculated for the assumed formula Ln(BTC) are given, which does not allow to quickly assess the chemical purity of the products.
- In the title, “Luminescent sensors” are mentioned, however, no sensor properties were studied in the work presented. I suggest changing this title from “luminescent sensors” to “luminescent properties”
Author Response
The Responses to Reviewers (Manuscript Inorganics-1391218)
Title: Nanostructure and Luminescent Properties of Bimetallic 2 Lanthanide Eu/Gd, Tb/Gd and Eu/Tb Coordination Polymers
Dear Reviewers,
Thank you for your comments for the manuscript Inorganics-1391218 by H. Brunckova et al. I have made some revisions by following your valuable suggestions.
Reviewer #2:
A: Thank you for comments and suggestions. The Introduction, results and conclusions were improved. These changes were incorporated in the Manuscript highlighted in green color. English was corrected.
Comments and Suggestions for Authors
The manuscript by Brunckova et al. reports the synthesis and luminescent properties of a series of isostructural homo- and heterobimetallic Eu, Gd, Tb metal-organic frameworks. The MOFs were prepared by the procedure previously reported by the same authors and were characterized by XPS, SEM and EDS methods.
I believe the manuscript fits the scope of the journal, however, some corrections are advisable:
- In the introduction part, the authors give some general information on MOFs and Ln-MOF in particular. However, they cite particular works ([1]-[8]), rather than more general review articles. I suggest including some references to recent reviews on MOFs and Ln-MOFs. The authors may find useful these two reviews: 10.1070/RCR5026 and 10.3390/ma13122699
A: The two reviews were included in the Manuscript.
Coordination polymers (CPs) are constructed of metal ions and bridging ligands that combine them into solid-state structures extending in one (1D), two (2D), or three dimensions (3D). Two- and three-dimensional CPs with potential voids are often designated to as metal-organic frameworks (MOFs) [1,2].
[1] Kuznetsova, A.; Matveevskaya, V.; Pavlov, D.; Yakunenkov, A.; Potapov, A. Coordination Polymers Based on Highly Emissive Ligands: Synthesis and Functional Properties. Materials 2020, 13, 2699-2766. https://doi.org/10.3390/ma13122699
[2] Kovalenko, K.A.; A S Potapov, A.S.; Fedin, V.P. Micro- and mesoporous metal-organic coordination polymers for separation of hydrocarbons, Rus. Chem. Rev. 2021, 90. https://doi.org/10.1070/RCR5026.
- In Table S1 elemental composition of Ln-MOFs are given, however, no values calculated for the assumed formula Ln(BTC) are given, which does not allow to quickly assess the chemical purity of the products.
A: The XPS study is known as technique for analyzing surface of compounds and does not prove that the structure of these LnBTC consists of a regular array of lanthanide ions surrounded by a BTC linker. The hydrogen element was not specified.
The values calculated for the assumed formula Ln(BTC)(DMF)2(H2O), {Ln[C6H3(CO2)3](C3H7NO)2(H2O), C15H19N2O9Ln are C7HxN2O7Eu, C11.6HxN2.8O9Gd, C12.4HxN3.5O9.4Tb, C10.8HxN1.4O7.6 Eu0.52Gd0.48, C14.1HxN2.6O10.6 Tb0.47Gd0.53 and C14.1HxN2.3O8.9 Eu0.47Tb0.53.
- In the title, “Luminescent sensors” are mentioned, however, no sensor properties were studied in the work presented. I suggest changing this title from “luminescent sensors” to “luminescent properties”
A: The title was changed.
Nanostructure and Luminescent Properties of Bimetallic Lanthanide Eu/Gd, Tb/Gd and Eu/Tb Coordination Polymers.
Yours sincerely,
Helena Brunckova
Institute of Materials Research Slovak Academy of Sciences
Watsonova 47, 040 01 Kosice, Slovakia
E mail: hbrunckova@saske.sk
Reviewer #1:
Comments and Suggestions for Authors
The article entitled “Nanostructure of Bimetallic Lanthanide Eu/Gd, Tb/Gd and Eu/Tb Metal-Organic Frameworks as Luminescent Sensors” by Brunckova et al., intends to present the synthesis and luminescence investigations for three “MOFs type” compounds with lanthanide (Eu,Gd,Tb) and other three “MOFs type” bimetallic compounds (Eu/Gd, Tb,Gd, Eu/Tb) with BTC used as ligand.
First of all, I disagree with the authors' assertion that the compounds provided in this manuscript are metal organic frameworks (MOFs). Even if the authors intended to characterize these lanthanide compounds by using XPS and XRD analysis, the data presented into the manuscript are inconsistent and has not enough accuracy.
The study requires more detailed investigations to propose these lanthanides compounds as metal-organic frameworks. The authors present the XRD powder of these compounds as similar with LnMOF, beeing isostructural compounds, see page 127, without a X-ray single crystal measurement which is of most importance to characterize MOFs structures. The XPS study is known as a technique for analyzing a material's surface and doesn't prove that the structure of these compounds consists of a regular array of lanthanide ions surrounded by a BTC linker. Even the XRD study is not correctly correlated with the literature.
A: Thank you for comments and suggestions. The Introduction, results and conclusions were improved. These changes were incorporated in the Manuscript highlighted in yelow color.
Major questions on the paper:
Line 2: The title is inadequate with the content of the manuscript. Again I should point out that not all mixtures with lanthanides and ligands lead to MOFs products and this is the case for this study too.
A:The title was changed to Nanostructure and Luminescent Properties of Bimetallic Lanthanide Eu/Gd, Tb/Gd and Eu/Tb Coordination Polymers.
Line 15: It is only general information about MOFs; should be in the introduction. Please reformulate to provide accurate information regarding the identified products.
A: The term coordination network solids can be seen as a compromise: IUPAC nomenclature can be adhered to even if coordination polymer is avoided. Metal-organic frameworks will thus be a subclass of coordination network solids, which in its turn is a subclass of coordination polymer. An alternative classification that avoids introducing new terms is to adopt a very broad inclusive definition of metal-organic framework as: “any system that forms a 2D or 3D network with carbon containing ligands bridging mononuclear, polynuclear or 1D coordination entities” [Ref x]. https://doi.org/10.1039/C2CE06488J
The sentence was reformulated. The study presents synthesis, structural and luminescence properties for lanthanide metal-organic frameworks (LnMOFs), which belong to a sub-class of coordination polymers.
Line 21-22: Please explain how the” nanostructure of mixed LnMOFs” could be related to luminescence propriety.
A: Luminescent nanomaterials, specifically assembled nano-architectures, demonstrate a better performance in terms of photoluminescence response together with customized microstructures and morphologies compared to bulk counterparts [Ref 2]. https://doi.org/10.3390/ma14164591 The properties and applications of luminescent materials are strongly dependent on their chemical composition, crystal structure, size and morphology. The Zn2GeO4:Mn2+ nanobundles exhibit broader diameters and larger particle size, which induces lower surface area and leads to fewer defects and stronger luminescence [Ref 3]. https://doi.org/10.1039/C8RA08636B
Line 35: the reference [3] is inappropriate.
A: The reference was changed.
Line 52: Has been established already in line 19 that LnMOFs are noticed as LnBTC; please avoid repetition.
A: Thank you for suggestion.The sentence was corrected.
Line 53: Please explain why BTC is classical??
A: Zhang et al. used term classical BTC in literature [Ref 4]. https://doi.org/10.1039/x0xx00000x BTC ligand is classical because it is used in various coordinating polymers with metallic Cu, Pb, Ni, Fe ions in addition to Ln ions. The solubility of H3BTC acid in pure water, isopropyl alcohol, isobutyl alcohol, methanol, ethanol, and ethylene glycol is good at the temperature range from 298 to 360 K.
Line 65: replace “Eu / GdMOF” with “Eu/GdMOF”;
A: The word was corrected.
Paragraph 96-109: There are many concerns about the preparation of these “MOFs”: the amount of reactants, the ratio between Ln:TBC not presented, isolation process, purification process, how/if was separate from the un-reacted reactants, the yields; elemental analysis, reproducibility. The preparation of these compounds is too important to be just simply presented as it is in this paper. The amount of reactants should be added, the isolation procedure should be described, the purification procedure should be presented, and a comparison with other methods and procedures already reported in the literature should be introduced.
A: The paragraph was corrected. The amount of reactants, the ratio between Ln:TBC, isolation process, purification process, the yields, reproducibility and comparison with other methods and procedures were supplemented.
LnMOFs were prepared via modified solvothermal synthesis [3,11,12] pursuant to the previous work [20]. Lanthanide(III) nitrate hydrate Ln(NO3)3·xH2O (1.0 mmol) and H3BTC (0.21 g, 1.0 mmol) were dissolved in the 30 mL mixture of DMF/H2O (1:1v/v) solvents (Ln = Eu, Gd, Tb) together with the modulator NaOAc (0.3 mmol). The preparation procedures for Eu. Gd and Tb lanthanide MOFs were same and were performed using different starting nitrates using Eu(NO3)3. 5H2O (0.443 g), Gd(NO3)3. 6H2O (0.448 g) and Tb(NO3)3. 5H2O (0.449 g). The ratio between Ln:TBC is 0.36. The three solutions of EuBTC (Eu-1), GdBTC (Gd-2) and TbBTC (Tb-3) were mixed at 25°C for 1 h and heated at 60°C for 48 h and then cooled to room temperature to give white (Eu-1 and Gd-2) and colorless (Tb-3) crystals. After synthesis, the products were isolated by centrifugation and washed several times with ethanol and water, respectively and then dried in air. The prepared Eu-1, Gd-2 and Tb-3 resulted with yield of 66% (0.319 g), 69% (0.270 g) and 70% (0.339 g), respectively without elemental analysis. The preparation of mixed bimetallic Eu0.5Gd0.5BTC (EuGd-4), Tb0.5Gd0.5BTC (TbGd-5) and Eu0.5Tb0.5BTC (EuTb-6) is the same to that for simple LnBTC, only pure Ln nitrate was exchanged via mixture of two corresponding nitrates. For preparation of EuGd-4 powder, the nitrate of Eu (0.223 g, 0.5 mmol) and Gd (0.224 g, 0.5 mmol) with BTC (0.21 g, 1.0 mmol) and NaOAc (0.03 g, 0.3 mmol) were dissolved in 30 mL DMF/H2O. For TbGd-5 and EuTb-6 syntheses was used Tb(NO3)3·6H2O (0.225 g, 0.5 mmol). The bimetallic LnMOF powders were formed after heating at 60°C for 48 h. The preparation procedures for other lanthanide MOFs were analogous. The experimental synthesis of each samples was repeated for three times. The syntheses of Tb-3 and TbGd-5 MOFs were performed without the NaOAc modulator for SEM and TEM analysis.
The comparison with other methods and procedures for solvothermal synthesis of LnBTC (Ln= Eu0.5/Gd0.5 or Tb0.5/Gd0.5) were used the selected lanthanide chloride salts, sodium trifluoroacetate (NaTFA), and BTC in ~1:0.9:0.6 Ln:TFA:BTC ratio with the solvents water and DMF [7] and ball milling preparation of Eu0.5/Gd0.5(BTC) or Tb0.5/Gd0.5(BTC) with the H3BTC and respective lanthanide carbonate hydrates Ln2(CO3)3·xH2O in 2:1 molar ratio. The other solvothermal synthesis of LnMOFs have been reported with various ligands as psa [15], H2OBA [16], acid H2FDA [17], H4L+Cl- [18] and NDC [19].
Also please point out if there have been performed several experiments for each compound (1-7) and provide more information regarding the reproducibility of the results of the produced samples. Reproducibility is essential for MOFs. Also please provide some information about their stability.
A: The experimental synthesis for each compound (1-6) was repeated for three times. The prepared Eu-1, Gd-2 and Tb-3 resulted with yield of 66% (0.319 g), 69% (0.270 g) and 70% (0.339 g), respectively without elemental analysis. EuGd-4, TbGd-5 and EuTb-6 resulted with yield of 65%. All the samples had similar thermal stability up to 540-600°C. The Figure S1 DSC/TG curves of LnMOFs (LnBTC) prepared by solvothermal synthesis was included in the manuscript.
How authors could explain why the conversion of the ligand is total ?? The unreacted compounds could be present in the synthesized compounds as Tb0.5Gd0.5-MoF or Eu0.5Tb0.5 MOF??
A: The solutions of LnBTC were mixed at 25°C for 1 h and heated at 60°C for 48 h, until complete dissolution. The products were isolated by centrifugation and washed several times with ethanol and water, respectively and then dried in air. We assume that the unreacted compounds could be present in the synthesized compounds as Tb0.5Gd0.5-MoF or Eu0.5Tb0.5 MOF in small quantities. Some unreacted residues of MOF compounds that were not involved in the polymerization may be dispersed in the polymer matrix.
Line 117: Please add the name of the EDX instrument.
A: Surface morphologies were characterized using scanning electron microscopy (SEM), (Auriga Compact, Carl Zeiss Germany) and high resolution transmission electron microscopy (TEM), (JEOL-JEM 2100F) and energy dispersive X-ray (EDS, Oxford Instruments X-max80 SDD detector) spectroscopy.
Paragraph 126-135: How can you explain the peaks at 1750 -1680 cm-1 from the Figure S1 (Supplementary information)?? The FTIR spectra are not explained in detail and the presence of some peaks could suggest the presence of BTC ligand, and to my opinion, the purity of the products was not proved.
A: The peaks at 1750 -1680 cm-1can be explain: In spectra, the characteristic bands of carboxyl group of BTC disappear (3090 and 1720 cm-1) and new peaks appear at 1568-1538 and 1386, cm-1, which belong to the stretching nas(COO-) and ns(COO-) vibrations of the carboxylic ions.
In spectrum of H3BTC acid, the characteristic bands of carboxyl group of BTC are at 3090, 1720, and 537 cm-1. In the spectra of MOFs, no band at 1720 cm-1 corresponding to the COOH groups is seen, designating the complete deprotonation of the carboxylic acid and coordination of COO- groups to the lanthanide centre. Peak at 1678 cm-1 belongs to ν (C=O) of DMF.
Paragraph 136-146: Here is the main concern regarding the purity and composition of these products. The authors cite their previous paper [20] indeed, but the data presented here are identical to that reported in ref. 20. It cannot be accepted this sentence ''almost identical peaks, which confirm their coincident crystal structure" reference [12,23], because in the figure S2 are a lot of missing data as follow: the 5-10 (2theta) interval; simulated patterns; and also the fingerprint of XPD patterns do not coincide with those from the cited reference[12,23].
The XPD patterns presented in ref[12, 23, 33] or simulated XPD from X-ray single crystal of Eu(BTC) - (CCDC 290771) are not fitting with XPD patterns presented in this manuscript and they must fit. This is a strong doubt that the structure of these products is not as presented in the article. The simulation patterns would prove the similarity of the signals, thus must be included in the paper.
A: The text was corrected. The simulation patterns were not performed.
Figure S2: Due to the fact that EuGd-4 covers the signals of compounds 5 and 6, please reduce the size of EuGd-4 signal.
A: The Figure S2 was corrected.
Line 140 The XRD patterns of Gd-2 is not ‘ín good agreement’ with ref [5] as was suggested, please change the reference.
A: The Ref [5] was changed to Ref [11].
Line 141: there is a mix of the results with the typical introduction text. Please rewrite it; the sentence is not correlated with the data reported here.
A: The sentence was deleted.
Line 142-146: There are only reported data that are not correlated with the new data (for introduction), again is mixing results with the introduction typical text. Please rewrite.
A: The text was rewrited.
The XRD diffractograms of samples are shown in Figure S2. The XRD peaks are in good compliance with previous report indicating that obtained MOFs are isostructural [20]. XRD diffractograms are in good agreement with patterns of Ln(BTC), which have tetragonal structure [11]. The Eu(BTC)(H2O)DMF, built with Eu3+ cation and the organic linker BTC via coordination bonding, is 3D framework with tetragonal structure [3,4,11]. XRD shows that Eu(BTC)(H2O)DMF structure is not changed using slight addition of NaOAc, but ever it will be transformed to monoclinic structure in the effect of major quantity of modulator agent [2,3].
Paragraph 149-176: Why it was decided to do the peak fitting XPS high resolution spectra of Eu 4d, Tb 4d and not 3d orbital?? Why the 3d orbitals presented in figure 1 are not deconvoluted?
A: I decided to presenting XPS high resolution spectra of Eu 4d, Tb 4d, because in available references are infrequent. The 3d orbitals presented in Figure 1 are not deconvoluted, because in the previous work [20] they are mentioned.
Eu+2 could be a surface defect?? Please explain clear the presence of both valences.
A: I think, that the Eu+2 is not surface defect. The high resolution Gd, Eu and Tb 3d was measured at HV 20 and HV 50 (mean resolution of the machine).
The presence of two valence states Eu3+/Eu2+ and Tb3+/Tb4+ in the corresponding samples was confirmed. It can also be caused by preparation of samples and drying in air.
Figure 1 – fitting (deconvolution) of the high-resolution spectra for each element;
A: The deconvolution of C 1s, N 1s and O 1s peaks was visible in Figure 3. Fitting of the high-resolution spectra of Eu, Gd aTb are in previous worwork [20].
Figure 2: move the figure in SI
A: The Figure 2 was inluded to SI as Figure S4.
Figure 3: In Figure 3 for C 1s are presented bonds C=O; in O 1s Eu-C-O and N 1s O-N-C-H whose presence is not justified; please explain their presence.
A: In Figure 3 for C 1s are presented bonds C=O; in O 1s, Eu-C-O; and N 1s O-N-C-H; caused by the presence of DMF and BTC.
The XPS results for C 1s and N 1s are in good agreement with the FTIR n(C=O) and d (O=C-N) from DMF coordinated, respectively.
Paragraph 203-214: If the rods of TbMOF are in the range 2-10µm that means 2000-100000 nm and that is not in concordance with Table 1. How do you explain that??
A: In Figure S4, the TbMOF morphology without and with modulator NaOAc is presented.
The TbMOF morphology is formed typical rods in the range 2 to 10 mm without modulator (Figure S4a [10]. In Figure S4b, needle-shaped nanocrystals of TbMOF after addition of modulator NaOAc are observed [13].
Figure 4 b presents the element At% for TbGd-5 that are not in concordance with table S2. The EDS proves for probe 4, show the composition for TbGd-5, 3.3 at 3.2, but in the Table S2 seems to be different (3.0 at 2.2). Please explain the discrepancies.
A: Figure 4a,b presents SEM/EDS ( C, O, N and Ln) morphology of TbGd-5 prepared by solvothermal synthesis without modulator NaOAc. In Table S2 is EDS analysis of TbGd-5 sample from STEM/EDS spectra of sample prepared using modulator. The elemental proportions on the surface of LnMOFs distinct from bulk content acquired using SEM/EDS analysis (C, O, Ln), while the values of carbon gained with the STEM rather higher (carbonized sample).
Also, the graphical representation released in Figure 4 (b and d) have data different than those presented in Table S2.
A: In Figure 4c,d (SEM/EDS) are data different to in Table S2 (STEM/EDS).
In the figures 4 and S4, the nitrogen is present. Please explain the presence of nitrogen. It could be from nitrate or DMF?
A: The presence of nitrogen is from DMF. I assume that nitrates have reacted with BTC.
Line 226: the sentence “The records acquired using TEM were fully alike and well agreed with the results gained from SEM” is not really sustained. In Figure 1 are presented TgGd-5 with rods at ~10-20 µm, and in SEM are 200 nm. As previously mentioned (line 203) the SEM and TEM are surface morphology methods and can not elucidate the “homogeneous partition” as was written line 245. Please reconsider the paragraph.
A: The Figure 4a,b presented TbGd-5 with rods at ~10-20 µm (prepared without modulator) and in Table 1 are the variation in sizes of nanorods obtained from the TEM images (Figure 5).
The records acquired using TEM were fully alike and well agreed with the results of samples prepared solvothermal synthesis with addition of modulator (NaOAc) gained from SEM.
The STEM and EDS mapping of elements in LnMOFs (Figure 6) elucidate the homogeneous partition of Ln, O and C elements through the surface of samples [32].
Figure 5: figure 5 d is not clear. Please explain in detail that figure?
A: The Figure present one rod which small nanorods (length 300-450 nm and width 50-100 nm) are incorporated.
In Figure 5d present one rod with length 850 nm and width 400 nm, which small nanorods (length 300-450 nm and width 50-100 nm) are incorporated.
Line 245: Please be consistent with STEM or TEM meaning. Please use the same terminology or describe the instrument in the instrumental part.
A: Scanning transmission electron microscope (STEM) is type of transmission electron microscope (TEM). I include Scanning transmission electron microscope (STEM) as type of transmission electron microscope (TEM).
Surface morphologies were characterized using scanning electron microscopy (SEM), (Auriga Compact, Carl Zeiss Germany) and high resolution transmission electron microscopy (TEM), (JEOL-JEM 2100F), Scanning transmission electron microscope (STEM) and energy dispersive X-ray (EDS, Oxford Energy TEM 250) spectroscopy.
Paragraph 256-298: Why was included in the manuscript the luminescence investigations only from probes 5,6, and 7, and not also from Eu-1 and Tb -3,? The excitation and emission spectra were recorded in solid-state? Please mention that. Have you recorded data also for BTC? Please introduce in the manuscript.
A: Luminescence investigations were not performed in samples Eu-1, Gd-2 and Tb-3. We have not recorded data for BTC.
The excitation and emission spectra of samples were recorded in solid-state.
Line 276: The characteristic peaks recorded in the luminescence spectra can be from the Ln+3 and from lanthanide nitrate. Please explain why you excluded the signal from lanthanide nitrate.
A: The lanthanide nitrate were not determined in LnMOFs, prepared by solvothermal synthesis using mixture of solvents EtOH/H2O [20]. XPS and SEM/EDS investigations confirmed no presence of nitrogen. Our samples were prepared by a similar procedure with the same amount of reactants with change in DMF/H2O solvents and the addition of a Modulator. We assume the absence of nitrates, which should react with H3BTC.
Line 298: Please move this sentence to the conclusion part.
A: The sentence was moved to the Conclusion part.
Table 2 can be moved to SI content.
A: Table 2 was moved to SI as Table S3.
Could you please add the thermogravimetric Analysis (TG/DTG) as you presented in the previous paper from the group [ref 20]
A: The thermogravimetric Analysis ( DSC/TG) was added as Figure 1.
Figure 1 DSC/TG curves and of LnMOF (Ln = Eu, Gd, Tb) prepared by solvothermal synthesis.
TG and DSC curves of LnMOFs (Eu-1, Gd-2, Tb-3) prepared by sovothermal synthesis using modulator (NaOc) are shown in Figure 1. The TG curves are similar, all of which display a two-step or three-step weight loss [3]. The initial weight loss starting at around 100°C up to 160°C observed for all samples can be ascribed to the loss of coordinated solvent molecules (DMF and H2O) [20]. From TG curves of Eu-1and Gd-2 shows two main steps of gradual weight loss process before 220°C, attributing to the release of H2O and two DMF molecules and 7.0 % in 220-410°C temperature range, corresponding to the loss of coordinated DMF molecules. Above 410°C, the complete framework sequentially begins to collapse upon further heating. All the samples had similar thermal stability up to 540-600°C. The weight loss at higher temperatures of 540-600°C is attributed to the thermal decomposition of LnBTC to Ln oxides. From DSC curves result that the end point of endo peak depends on chemical component of lanthanides (Eu-1: 620°C, Gd-2: 658°C and Tb-3: 673°C).
The manuscript needs more and careful checks of the characterization of the product.
A: Thank you for comments and suggestions. I have made careful checks and corrections.
The Figures and Tables in Response to Reviewer comments are marked as in the reviewer comments of uncorrected Manuscript. The references I revised Manuscript were corrected.
Figure Caption: in revised Manuscript Inorganics-1391218)
Figure 1 DSC/TG curves of LnMOF (Ln = Eu, Gd, Tb) prepared by solvothermal synthesis.
Figure 2 HR XPS spectra of Eu 3d, Gd 3d, Tb 3d, C 1s, O 1s and N 1s of LnBTC (Eu-1, Gd-2, Tb-3, EuGd-4, TbGd-5 and EuTb-6) powders.
Figure 3 HR XPS spectra of C 1s, O 1s and N 1s and curve fitted peaks for EuBTC and Eu0.5Tb0.5BTC powders.
Figure 4 SEM morphology and TEM images (in insert) of LnMOF powders prepared by solvothermal synthesis a)TbGd-5 (DMF/H2O), c) EuTb-6 (DMF/H2ONaOAc) and EDS spectra of b) Tb0.5Gd0.5BTC and d) Eu0.5Tb0.5BTC.
Figure 5 TEM images of LnMOF powders a) Eu-1, b) Gd-2, c) Tb-3, d) EuGd-4, e) TbGd-5 and f) EuTb-6.
Table 1 The variation in sizes of nanorods obtained from the TEM images (Figure 5).
Figure 6 STEM images and EDS elemental mapping of LnMOF powders a) Tb-3, b) EuGd-4, c) TbGd-5 and d) EuTb-6 samples.
Figure 7 Excitation spectra of LnMOF samples a) EuTb-6, TbGd-5 (λem = 543 nm) and b) EuTb-6, EuGd-4 (λem = 617 nm).
Figure 8 a) Emission spectra and b) CIE chromaticity diagrams of bimetallic lanthanide MOFs excited at 300 nm marked a) EuTb-6, b) TbGd-5 and c) EuGd-4.
Figure Caption in SI: in revised Manuscript Inorganics-1391218)
Figure S1. FTIR spectra of LnMOFs prepared by solvothermal synthesis.
Figure S2. XRD patterns of LnMOFs prepared by solvothermal synthesis.
Figure S3. XPS survey spectra of LnMOFs prepared by solvothermal synthesis.
Table S1. XPS elemental atomic % of LnMOF samples.
Figure S4 HR XPS spectra of Eu 4d and Tb 4d of EuBTC, TbBTC and Eu0.5Tb0.5BTC samples.
Figure S5. SEM morphology and TEM images (in insert) of TbMOF powders prepared by solvothermal synthesis a) (DMF/H2O), c) (DMF/H2O/NaOAc) and b,d) EDS spectra of TbBTC.
Table S2 EDS analysis of Tb-3, EuGd-4, TbGd-5 and EuTb-6 samples from TEM/EDS spectra.
Table S3 The applications of luminescent bimetallic LnMOFs.
Yours sincerely,
Helena Brunckova
Institute of Materials Research Slovak Academy of Sciences
Watsonova 47, 040 01 Kosice, Slovakia
E mail: hbrunckova@saske.sk

Round 2
Reviewer 1 Report
The article “Nanostructure and Luminescent Properties of Bimetallic Lanthanide Eu/Gd, Tb/Gd and Eu/Tb Coordination Polymers” has been amended by the authors according to the reviewer's suggestion and comments and deserves publication in present form.
Author Response
Editor-in-chief: Inorganics MDPI
Prof. Dr. Claudio Pettinari
Inorganic Chemistry Unit, School of Pharmacy-ICCOM-CNR Camerino, University of Camerino, via S. Agostino 1, 62032 Camerino, Italy
Miss Paula-Iuliana Andone
Section Managing Editor, MDPI Romania. Avram Iancu 454, 407280 Floresti, Cluj, Romania
Kosice, October 12, 2021
Dear Professor Claudio Pettinari and Miss Paula-Iuliana Andone,
enclosing you will find our revised manuscript (Inorganics-1391218) "Nanostructure and Luminescent Properties of Bimetallic Lanthanide Eu/Gd, Tb/Gd and Eu/Tb Coordination Polymers" by Helena Brunckova, Erika Mudra, Lucas Rocha, Eduardo Nassar, Willian Nascimento, Hristo Kolev, Maksym Lisnichuk, Alexandra Kovalcikova, Zuzana Molcanova, Magdalena Streckova and Lubomir Medvecky. The revision made according to the Academic Editor Notes. We confirm that this manuscript has not been submitted to another journal and present completely novel results of new bimetallic mixed LnMOFs luminescent sensors. We hope this article will prove to be eligible for publication in Inorganics.
The responses to the Editor we include in the Cover Letter. Thank you for attention.
Yours sincerely,
Helena Brunckova
Institute of Materials Research, Slovak Academy of Sciences
040 01 Kosice, Watsonova 47, Slovakia
The Responses to the Editor (Manuscript Inorganics-1391218)
Title: Nanostructure and Luminescent Properties of Bimetallic Lanthanide Eu/Gd, Tb/Gd and Eu/Tb Coordination Polymers
Dear Editor,
Thank you for your comments for the manuscript Inorganics-1391218 by H. Brunckova et al. I have made some revisions by following your valuable suggestions.
Academic Editor Notes
Thank you for your diligent revision of this manuscript. The editorial team has just a few minor things for you to revise, then this manuscript will be accepted.
Ref 1 and 2 are not suitable general MOF reference. Basically the first phrases reiterate the IUPAC definition that then needs to be cited. Some more general MOF references should also be added, recent books for example, but the added ref 1 and 2 can be kept.
A: I included the References [1-3] in Introduction part.
Coordination polymers (CPs) are constructed of metal ions and bridging ligands that combine them into solid-state structures extending in one (1D), two (2D), or three dimensions (3D). Two- and three-dimensional CPs with potential voids are often designated to as metal-organic frameworks (MOFs) [1-5].
[1] Batten, S.R.; Champness, N.R. ; Chen, X.M.; Garcia-Martinez, J.; Kitagawa, S. ; Öhrström, L.; ’Keeffe, M.; Suh, M.P.; Reedijk, J. Terminology of metal-organic frameworks and coordination polymers (IUPAC Recommendations 2013)*. Pure Appl. Chem. 2013, 85, 1715-1724. [http://dx.doi.org/10.1351/PAC-REC-12-11-20]
[2] Peedikakkal A.M.P., Adarsh N.N. Coordination Polymers. In Book Porous Coordination Polymers. In: Jafar Mazumder M., Sheardown H., Al-Ahmed A. (eds) Functional Polymers. Polymers and Polymeric Composites: A Reference Series. Springer, Cham. pp. 2-6.
[https://doi.org/10.1007/978-3-319-92067-2_5-1]
[3] Lee, J.S.M.; Otake, K.; Kitagawa, S. Transport properties in porous coordination polymers. Coordination Chem. Rev. 2020, 421, 213447. https://doi.org/10.1016/j.ccr.2020.213447
Formulas such as Eu(BTC)(H2O)(DMF) should, according to IUPAC, use small letters for abbreviations and square brackets encircling the coordination unit, i.e. [Eu(btc)(H2O)(dmf)] please revise.
A: I corrected the formulas in the Manuscript.
You state ”The XRD peaks are in good compliance with previous report indicating that obtained…” and we have no reason to doubt this, however, please provide a graph in the SI figure S2 that shows that this is the case.
A: The [La(btc)] patterns were added in Figure S2 [25,28]. I added the Ref. [28].
The XRD diffractograms of samples are shown in Figure S2. The XRD peaks are in good compliance with previous report indicating that obtained MOFs are isostructural [25]. XRD diffractograms are in good agreement with patterns of [Ln(btc)], which have tetragonal structure as [La(btc)] simulated [28]. The diffraction patterns indicate the formation of [Ln(btc)], (Ln = Eu, Gd, Tb, Eu0.5Gd0.5, Tb0.5Gd0.5 and Eu0.5Tb0.5) in 2θ of 10.5°, 18.3°, 20.3°, 27.5° and 32° the most important peaks [25,28]. The other peaks also match the [La(btc)] structure.
[28] Song, K.; Yu, H.; Zhang, J.; Bai, Y.; Guan, Y.; Yu, J.; Guo, L. Rosebengal-Loaded Nanoporous Structure Based on Rare Earth Metal-Organic-Framework: Synthesis, Characterization and Photophysical Performance. Crystals 2020, 10, 185–199. [https://doi.org/10.3390/cryst10030185]
Figure S2. XRD patterns of LnMOFs prepared by solvothermal synthesis.
Some details:
“imported from Sigma-Aldrich” change to “purchased from Sigma-Aldrich”
A: The sentence was corrected in Experimental part.
All chemical agents and dissolvents have been purchased from Sigma-Aldrich with analytical grade and applied without another purgation.
”XPS was corroborated the coordination impact among europium, gadolinium and terbium ions and BTC ligand.” change to ”XPS corroborated the coordination impact among europium, gadolinium and terbium ions and BTC ligand.”
A: The sentence was corrected in Conclusions part.
XPS corroborated the coordination impact among europium, gadolinium and terbium ions and BTC ligand.
”For TbGd-MOF, the CIE coordinates were presenting emission in green region of chromaticity diagram. ” change to ”For TbGd-MOF, the CIE coordinates show emission in green region of chromaticity diagram.”
A: The sentence was corrected in Conclusions part.
For TbGd-MOF, the CIE coordinates show emission in green region of chromaticity diagram.
Yours sincerely,
Helena Brunckova
Institute of Materials Research Slovak Academy of Sciences
Watsonova 47, 040 01 Kosice, Slovakia
E mail: hbrunckova@saske.sk
This manuscript is a resubmission of an earlier submission. The following is a list of the peer review reports and author responses from that submission.
Round 1
Reviewer 1 Report
In this contribution, three bimetallic lanthanide MOFs were prepared, and their XPS, SEM, TEM, and luminescence spectra were measured and analyzed. The work was carefully done, but all the experimental results were routine, and no new or important outcome was provided. In addition, the LnBTC system has been largely investigated, it is not convincible to claim this is design of the bimetallic lanthanide MOFs. Therefore, in my opinion, this contribution is not a good fit for Nanomaterials.
Reviewer 2 Report
The manuscript by Helena Brunckova and co-workers very nicely explained the synthesis, characterization of mixed LnMOFs using benzene-1,3,5-tricarboxylate and bimetallic lanthanides (EuGd, TbGd and EuTb) followed by exploring their luminescent properties. It can be acceptable after few minor corrections.
- In page 2, line 65, there is a typographical error ‘he Eu/Tb’ should be ‘The Eu/Tb’.
- In Page 3, at the start of line 98, ‘solved’ would be ‘dissolved’.
- In page 7, in discussion 3.3, citation of Figure 4b should be 4c (line 212) and EDS spectra in Figures 4c would be 4b (line 213).
- It would be really nice if authors could provide the DLS of all six LnMOFs or should compare the variation in sizes of all nanorods obtained from the TEM images.
Reviewer 3 Report
In his work, although the chemical content, binding structures and morphologies on the surface of LnMOF series (Ln = Eu, Gd, Tb) and mixed (Ln = Eu0.5Gd0.5, Tb0.5Gd0.5 and Eu0.5Tb0.5) were characterized, this research lacks the novelty. Also, the authors indicated that "The mixed LnMOFs are very promising UV ligth sensors, due to their very high excitation bands and intense emissions. ", but no luminescent sensor applications were investigated, and no data reported to support "their very high excitation bands and intense emissions". So, I don't recommend it will be published in Nanomaterials.